

# HETEAC-Flex: An optimal estimation method for aerosol typing based on lidar-derived intensive optical properties

Athena Augusta Floutsi[1], Holger Baars[1], and Ulla Wandinger[1]

[1]Leibniz Institute for Tropospheric Research (TROPOS), Leipzig, Germany

**Correspondence:** Athena Augusta Floutsi (floutsi@tropos.de)

**Abstract.** This study introduces a novel methodology for the characterization of atmospheric aerosol based on lidar-derived intensive optical properties. The proposed aerosol-typing scheme is based on the optimal estimation method (OEM) and allows the identification of up to four different aerosol components of an aerosol mixture as well as the quantification of their contribution to the aerosol mixture in terms of relative volume. The four aerosol components considered in this typing scheme are associated with the most commonly observed aerosol particles in nature and are assumed to be physically separated from each other and, therefore, can create external mixtures. Two components represent absorbing and less absorbing fine-mode particles and the other two spherical and non-spherical coarse-mode particles. These components reflect adequately the most frequently observed aerosol types in the atmosphere: combustion- and pollution-related aerosol, sea salt and desert dust, respectively. In addition, to consolidate the calibration and validation efforts for the upcoming EarthCARE mission, the typing scheme proposed here is in accordance with the Hybrid End-To-End Aerosol Classification (HETEAC) model of EarthCARE. The lidar-derived optical parameters used in this typing scheme are the lidar ratio and the particle linear depolarization ratio at two distinct wavelengths (355 and 532 nm), the backscatter-related color ratio for the wavelength pair of 532/1064 nm and the extinction-related Ångström exponent for the wavelength pair of 355/532 nm. These intensive optical properties can be combined in different ways making the methodology flexible, allowing thus its application to lidar systems with different configurations (e.g., single wavelength or multiwavelength, Raman, high-spectral-resolution). The typing scheme was therefore named HETEAC-Flex, due to its compatibility with EarthCARE's HETEAC and methodological flexibility. The functionality of the typing scheme is demonstrated by its application to three case studies.

## 1 Introduction

Lidars in space have advanced our knowledge on aerosol, clouds and their interactions starting with the pioneering LITE (Lidar In-space Technology Experiment) instrument inside the payload bay of the Space Shuttle Discovery in 1994 (McCormick et al., 1993; Winker et al., 1996). CALIOP (Cloud-Aerosol Lidar with Orthogonal Polarization) – a direct descendant of LITE and the first polarization lidar in space – onboard NASA's (National Aeronautics and Space Administration) CALIPSO (Cloud-Aerosol Lidar and Infrared Pathfinder Satellite Observations) satellite has been acquiring global long-term atmospheric measurements since 2006 (Winker et al., 2009). CALIOP has measured vertical profiles of attenuated backscatter at visible (532 nm) and near-infrared (1064 nm) wavelengths, along with the linear depolarization ratio at 532 nm. However, CALIOP as an elastic-



backscatter lidar is not able to perform direct extinction measurements and to enable the retrieval of the backscatter and extinction coefficients from the attenuated backscatter signals, the lidar (extinction-to-backscatter) ratio needs to be assumed. Since the lidar ratio depends on the aerosol types present in the atmosphere, an aerosol-typing scheme was developed for CALIPSO (Omar et al., 2005, 2009; Kim et al., 2018; Tackett et al., 2023). CALIPSO's typing algorithm is able to classify

and assign typical lidar-ratio values to eleven different aerosol types, which can be found in the troposphere and stratosphere (Kim et al., 2018; Tackett et al., 2023). It becomes clear that the goodness of the extinction retrievals is always dependent on this typing scheme, even though several quality control procedures are in place (Winker et al., 2009).

Aeolus was the first spaceborne Doppler wind lidar (Stoffelen et al., 2006), and it was launched by the European Space Agency (ESA) in 2018. The mission was equipped with a 355-nm high-spectral-resolution lidar (HSRL), the Atmospheric

Laser Doppler Instrument (ALADIN). ALADIN acquired one-directional horizontal tropospheric and stratospheric wind profiles (mainly west-east), aiming to improve weather forecast, advance atmospheric dynamics research and evaluate climate models (Stoffelen et al., 2006; Straume et al., 2020). ALADIN's ability to measure extinction coefficients directly via the HSRL technique allowed the retrieval of aerosol and cloud optical properties as spin-off products (Ansmann et al., 2007; Flamant et al., 2008; Straume et al., 2020; Flament et al., 2021; Ehlers et al., 2022). The spin-off products include next to the

particle backscatter and the extinction coefficients also the particle lidar ratio at 355 nm. Even though preliminary, due to the ongoing algorithm improvements and quality assurance updates, first validation activities with ground-based lidar measurements showed promising results (Baars et al., 2021; Abril-Gago et al., 2022; Gkikas et al., 2023). With respect to aerosol typing, Aeolus had a drawback since it emitted circular-polarized light but detected the co-polar component of the backscattered light only. Consequently, part of the signal got lost in the case of depolarization by particles such as mineral dust, volcanic

ash, or ice crystals. Due to this signal loss, the backscatter coefficient is underestimated while the particle-specific lidar ratio is overestimated. This effect imposes challenges for aerosol typing.

The most recent lidar mission in space is the Atmospheric Environment Monitoring Satellite (AEMS), which is equipped with the Aerosol and Carbon dioxide Detection Lidar (ACDL), and it was launched on April 2022 by the China National Space Administration (CNSA; Han et al., 2018; Liu et al., 2019; Ke et al., 2022). ACDL is also an HSRL lidar and the first carbon

dioxide detection lidar in space. Along with the columnar concentration of carbon dioxide, ACDL acquires the vertical profiles of aerosols and clouds, to assess their impact on climate change and air quality.

The next lidar in space will be onboard the Earth Clouds, Aerosol and Radiation Explorer (EarthCARE) joint mission of ESA and the Japanese Aerospace Exploration Agency (JAXA), scheduled for launch in 2024. The ATmospheric LIDar (ATLID) will provide vertically resolved global measurements of the Earth's atmosphere (Illingworth et al., 2015). ATLID is a 355-nm HSRL

(Wehr et al., 2023) that will directly measure extinction and backscatter coefficients and, hence, the lidar ratio. Furthermore, ATLID will deliver the linear depolarization ratio of the atmospheric particles – a parameter that is ideal for aerosol-typing purposes (Mamouri and Ansmann, 2014; Illingworth et al., 2015; do Carmo et al., 2021; Floutsi et al., 2023). One of the major goals of the mission is radiative closure, which will be approached synergistically with EarthCARE's payload, which in addition to ATLID consists of a Cloud Profiling Radar (CPR), a Multi-Spectral Imager (MSI) and a Broad-Band Radiometer

(BBR; Illingworth et al., 2015; Wehr et al., 2023). An important prerequisite for achieving this goal is a proper aerosol-typing



scheme, which enables the calculation of the aerosol's radiative properties. For this purpose, the Hybrid End-To-End Aerosol Classification (HETEAC) model has been developed (Wandinger et al., 2023a). As the name indicates, the HETEAC model delivers the required theoretical description of aerosol microphysics that is consistent with experimentally derived optical properties (Floutsi et al., 2023, hybrid approach) in order to close the loop from observations and aerosol microphysics to
radiative properties (end-to-end approach).

In HETEAC, the aerosol types observed in nature are projected as a composition of four basic aerosol components. These components comprise two fine modes, one strongly absorbing and one weakly absorbing, and two coarse modes, one with spherical particles and one with non-spherical particles. A mono-modal particle size distribution and a wavelength-dependent complex refractive index are assigned to each of these components to obtain their microphysical description. The approach
has been adapted from ESA's Climate Change Initiative (CCI) project Aerosol_cci (Holzer-Popp et al., 2013). The parameters of the size distribution are global mean values obtained from the Aerosol Robotic Network (AERONET, Holben et al., 1998), which are considered typical for the aerosol components. The refractive indices are taken from different databases and selected such that the observations can be best reproduced with the microphysical particle model (for more details refer to Wandinger et al., 2023a). To describe the scattering of the non-spherical particles, the spheroid distribution of Koepke et al. (2015) was
chosen to best reproduce the observations. To account for aerosol mixtures of two or more modes, a multimodal representation is achieved by mixing rules. Each component has specific scattering properties per unit particle volume, which are used, in combination with the relative volume contribution of each component, to derive the optical properties of the aerosol mixture. This procedure results in look-up tables (LUT) of the optical and radiative properties for the different mixing ratios of the aerosol modes at specific wavelengths (Wandinger et al., 2023b). Once EarthCARE is in orbit, HETEAC will be used to reveal
the mixing ratio of the four different aerosol components from ATLID's measurements.

With HETEAC and a synergistic combination of the HSRL and MSI measurements, aerosol classification and quantification of the radiative impact will be available at global and regional scales. As for every satellite mission, ground-based remote-sensing measurements are essential for EarthCARE's product and algorithm validation. To facilitate the validation activities and with a specific focus on aerosol classification and radiative impact quantification, an aerosol-typing methodology appli-
cable to ground-based and spaceborne lidar systems has been developed and is presented in this study. The methodology is based on the optimal estimation method (Rodgers, 2000) and allows the identification of aerosol mixtures consisting of the aforementioned four different aerosol components from lidar measurements. The methodology is rather flexible, allowing thus its application to lidar systems at different wavelengths (e.g., 532 nm) and with different configurations (e.g., single wavelength or multiwavelength, Raman or HSRL). Microphysical and optical properties of the predefined aerosol components are in ac-
cordance with HETEAC, thus permitting direct comparisons, algorithm harmonization and further support to the validation activities for EarthCARE. Because of the flexibility of the methodology along with its compatibility with HETEAC, the name HETEAC-Flex has been chosen aerosol-typing scheme.

The paper is structured as follows. At the beginning (Sect. 2), a short introduction to the optimal estimation method is presented along with a comprehensive description of the aerosol-typing methodology and its main constituents. Then, the
aerosol-typing scheme is applied to three case studies in Sect. 3. Finally, the paper closes with the conclusions and outlook.



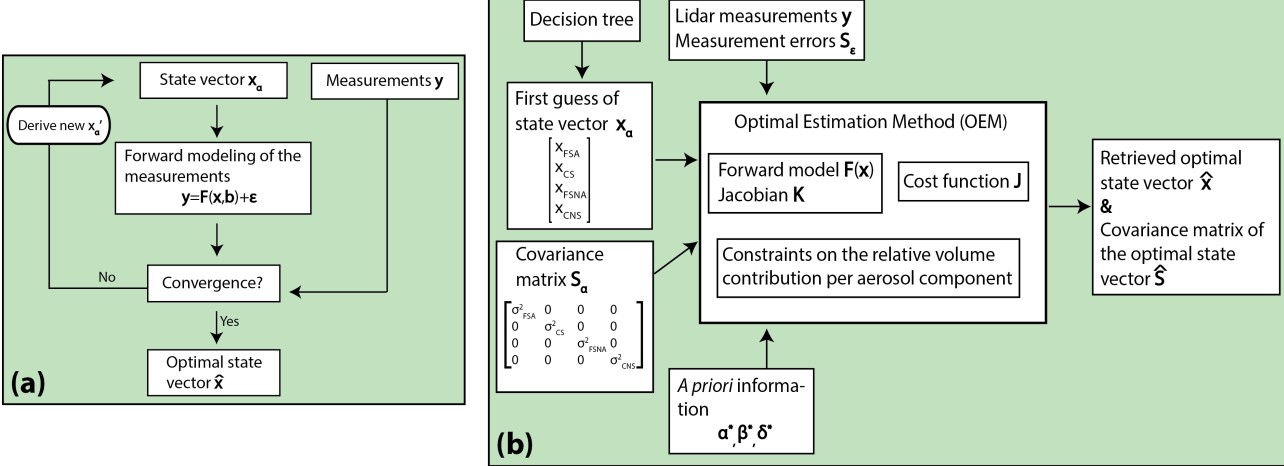

**Figure 1.** (a) Generalized concept of the optimal estimation method and (b) detailed illustration of the workflow of HETEAC-Flex.

## 2 Retrieval methodology

### 2.1 Overview

The optimal estimation method (OEM) is a nonlinear regression scheme applied to determine the statistically most-likely conditions to produce a given measurement, weighted against a priori knowledge of the system under investigation. A brief

overview of the retrieval framework is given in Fig. 1a. In basic terms, the process starts with an initial guess of the state vector ($\boldsymbol{x}_\alpha$, here the relative volume contribution per aerosol component), which is iteratively modified such that the modeled intensive optical properties match those observed by the lidar ($\boldsymbol{y}$), resulting in the most probable estimated state ($\widehat{\boldsymbol{x}}$). The problem is ill-posed, since several different states may produce the same measurements and, therefore, a priori information is needed to constrain the state space.

As outlined in Rodgers (2000), optimal estimation solves the inverse problem

$$\boldsymbol{y} = \boldsymbol{F}(\boldsymbol{x}, \boldsymbol{b}) + \boldsymbol{\epsilon}, \tag{1}$$

where $\boldsymbol{y}$ is a column vector describing the measurements, $\boldsymbol{\epsilon}$ is the corresponding noise (error) of these measurements and $\boldsymbol{F}(\boldsymbol{x}, \boldsymbol{b})$ the forward model that translates a state of the instrument and atmosphere, summarised by unknown parameters ($\boldsymbol{x}$) and known parameters ($\boldsymbol{b}$), into a simulated measurement. The probability that the system has a state ($\boldsymbol{x}$), given the measurements ($\boldsymbol{y}$), can

be found by approximating the probability density function ($P$) for all quantities as Gaussian, and using Bayes' theorem:

$$-2\ln P(\boldsymbol{x}|\boldsymbol{y}) = [\boldsymbol{y} - \boldsymbol{F}(\boldsymbol{x}, \boldsymbol{b})]^T \boldsymbol{S}_\epsilon^{-1} [\boldsymbol{y} - \boldsymbol{F}(\boldsymbol{x}, \boldsymbol{b})] + [\boldsymbol{x} - \boldsymbol{x}_\alpha]^T \boldsymbol{S}_\alpha^{-1} [\boldsymbol{x} - \boldsymbol{x}_\alpha], \tag{2}$$

where the covariance matrix $\boldsymbol{S}_\epsilon$ describes the measurement errors and $\boldsymbol{x}_\alpha$ the initial guess of the state (also referred to as the a priori state). The uncertainty in that expectation for the initial guess of the state is described by the a priori covariance $\boldsymbol{S}_\alpha$.





The quantity $-2\ln P(\boldsymbol{x}|\boldsymbol{y})$ is hereafter referred to as the cost, as it measures the goodness of fit for a solution. The iterational
process converges where the cost reaches a minimum, and that is the most probable state or the so-called optimal solution $\widehat{\boldsymbol{x}}$.
Convergence is evaluated by the smallness of the reduction of the cost function (see Sect. 2.5). Typically, the process converges
within 30 iterations and if not, then it fails to converge and, consequently, there is no optimal solution.

A detailed overview of the main elements of HETEAC-Flex is presented in Fig. 1b and in the following sections (Sect. 2.2–
2.6). Readers may refer to Rodgers (2000) for an in-depth explanation of the optimal estimation methodology and to Maahn
et al. (2020) for a comprehensive overview. For clarity, the notation of Rodgers (2000) is followed throughout this study.

## 2.2    State and measurement vectors

The quantities to be retrieved, which are the relative contributions of the aerosol components (to the total aerosol volume)
present in an aerosol mixture, are represented by the state vector $\boldsymbol{x}$. Similarly to HETEAC, every aerosol mixture is assumed
to consist of a maximum of four basic aerosol components, two fine modes (spherical absorbing and spherical non-absorbing;
FSA and FSNA, respectively) and two coarse modes (spherical and non-spherical; CS and CNS, respectively). Therefore, the
state vector consists of four dependent and continuous (in the interval [0, 1]) variables. The basic aerosol components enable
the estimation of several intensive optical properties (see Sect. 2.4). More details on the aerosol components and their optical
and microphysical properties are provided in Sect. 2.3.

The initial guess of the state vector $\boldsymbol{x}_\alpha$, along with its covariance matrix $\boldsymbol{S}_\alpha$, is needed to start the iterational process.
In the OEM, a priori information is used to regulate the ill-posed retrieval problem, making use of existing knowledge of the
atmosphere and making the solution a physically meaningful result. The retrieval is thus sensitive to the choice of $\boldsymbol{x}_\alpha$, which for
this retrieval scheme is the output of a decision tree (see Appendix A), while the setting of $\boldsymbol{S}_\alpha$, which describes the estimated
uncertainty of the initial-guess state vector elements as well as the correlation between the state vector elements, is a very
controversial part of the OEM, since it might constrain the solution space inappropriately (by forming a subspace in which
the solution must lie, with the correlations rejecting unrealistic solutions). In this study and for the given nature of $\boldsymbol{x}$, there
is no manner in which an exact covariance $\boldsymbol{S}_\alpha$ can be derived (e.g., based on existing observations as in Foth and Pospichal
(2017)) and hence, it is set in a parametric way. The variances of the aerosol components are given with some margin with
respect to the true values expected (Floutsi et al., 2023), and the covariances (non-diagonal elements) are set to zero, since
no relationship between the elements of the state vector is observed in nature, i.e., every basic aerosol component can coexist
independently from the others and there is no correlation between them (e.g., the existence of absorbing particles does not
prohibit or favor the existence of non-absorbing particles etc.). However, setting $\boldsymbol{S}_\alpha$ to a purely diagonal matrix translates to a
stronger regularisation (Rodgers, 2000).

The quantities actually measured and used to retrieve the state vector $\boldsymbol{x}$ are represented by the measurement vector $\boldsymbol{y}$. This
vector includes the intensive optical properties of the aerosol layer of interest, measured by the lidar. To be more specific, the
properties considered in this retrieval scheme are (in the order of appearance in the vector): the particle linear depolarization
ratio at 355 nm, the lidar ratio at 355 nm, the extinction-related Ångström exponent (for the wavelength pair of 355/532 nm), the
particle linear depolarization ratio at 532 nm, the lidar ratio at 532 nm and the backscatter-related color ratio (for the wavelength



**Table 1.** Microphysical properties and shape representation of the four basic aerosol components used to simulate multimodal particle distributions in HETEAC (Wandinger et al., 2023a) and HETEAC-Flex. $r_{\text{eff}}$ is the effective radius, $r_{0,\text{N}}$ and $r_{0,\text{V}}$ are the mode radii of the lognormal number and volume size distributions, respectively, $\sigma$ is the mode width (variance) and $m_{\text{R}}$ and $m_{\text{I}}$ are the real and imaginary part of the refractive index, respectively, at 355 and 532 nm.

| Property | Fine mode absorbing | Fine mode non-absorbing | Coarse mode spherical | Coarse mode non-spherical |
|---|---|---|---|---|
| $r_{\text{eff}}$ ($\mu$m) | 0.14 | 0.14 | 1.94 | 1.94 |
| $r_{0,\text{N}}$ ($\mu$m) | 0.07 | 0.07 | 0.788 | 0.788 |
| $r_{0,\text{V}}$ ($\mu$m) | 0.1626 | 0.1626 | 2.32 | 2.32 |
| $\sigma$ | 0.53 | 0.53 | 0.6 | 0.6 |
| $m_{\text{R}}$ (355 nm) | 1.50 | 1.45 | 1.37 | 1.54 |
| $m_{\text{R}}$ (532 nm) | 1.50 | 1.44 | 1.36 | 1.53 |
| $m_{\text{I}}$ (355 nm) | 4.3e−2 | 1.0e−3 | 4.0e−8 | 6.0e−3 |
| $m_{\text{I}}$ (532 nm) | 4.3e−2 | 1.0e−3 | 4.0e−9 | 3.0e−3 |
| Shape representation | Spherical | Spherical | Spherical | Spheroidal |

pair of 532/1064 nm). These intensive optical properties were chosen due to their high typing discrimination power (Burton et al., 2012). It should be noted that the measurement vector $\boldsymbol{y}$ can contain different combinations of the aforementioned properties, depending on the lidar capabilities and measurement availability (e.g., in the case of EarthCARE only the 355 nm particle linear depolarization ratio and lidar ratio will appear in the vector). Since measurements are made to a finite accuracy, the corresponding measurement errors are included in the diagonal matrix $\boldsymbol{S}_\epsilon$.

## 2.3 A priori aerosol components

The microphysical and optical properties of the four basic aerosol components considered in this retrieval scheme (i.e., FSA, CS, FSNA and CNS) are utilized as a priori information, which facilitates the construction of the forward model (discussed in Sect. 2.4). To ensure consistency between this typing scheme and HETEAC, the aerosol components along with their microphysical properties are the same as the ones defined in HETEAC (Wandinger et al., 2023a). The microphysical properties for the four aerosol components are summarized in Table 1. The effective radius ($r_{\text{eff}}$) is set to 0.14 $\mu$m for the fine-mode particles and to 1.94 $\mu$m for the coarse-mode particles. Together with the mode radius for the number and volume size distributions ($r_{0,\text{N}}$ and $r_{0,\text{V}}$, respectively), these quantities reflect the differences in the size of the particles. The real part of the refractive index at 355 nm ($m_{\text{R}}$) is highest for the CNS component, slightly lower and comparable for the two fine-mode aerosol components (FSA and FSNA) and lowest for the CS component. On the other hand, the imaginary part of the refractive index at 355 nm ($m_{\text{I}}$) is highest for the FSA component, followed by the CNS and FSNA components, and finally the CS component. The refractive index reflects the chemical composition of the aerosol particles.



**Table 2.** Optical properties of the four basic aerosol components at two wavelengths (355 and 532 nm). The extinction and backscatter coefficients per unit volume are abbreviated as $\alpha^*$ and $\beta^*$, respectively, the particle linear depolarization ratio as $\delta$ and the lidar ratio as $S$. $\alpha^*$ and $\beta^*$ are calculated for a unit particle volume of 1 $\mu m^3 cm^{-3}$.

| Aerosol component | $\alpha^* \times 10^{-12}$ (Mm$^{-1}$) | | $\beta^* \times 10^{-12}$ (Mm$^{-1}$sr$^{-1}$) | | $\delta$ | | $S$ (sr) | |
|---|---|---|---|---|---|---|---|---|
| | 355 | 532 | 355 | 532 | 355 | 532 | 355 | 532 |
| FSA | 10.7 | 6.45 | 0.09 | 0.07 | 0.024 | 0.024 | 117.3 | 93.8 |
| CS | 0.88 | 0.94 | 0.051 | 0.049 | 0.015 | 0.015 | 17.4 | 19.2 |
| FSNA | 9.61 | 5.03 | 0.16 | 0.08 | 0.033 | 0.033 | 60.9 | 59.3 |
| CNS (Saharan) | 0.93 | 0.97 | 0.016 | 0.018 | 0.24 | 0.33 | 57.9 | 55.0 |
| CNS (Asian) | 0.93 | 0.97 | 0.022 | 0.024 | 0.25 | 0.28 | 43.3 | 40.0 |

The scattering properties per unit particle volume (e.g., 1 $\mu m^3 cm^{-3}$) of the four different aerosol components that are used directly in HETEAC-Flex (see Eq. (3)–(6)) as a priori information are summarized in Table 2. For HETEAC, these properties have been calculated with a Mie scattering model for spherical particles (FSA, FSNA, CS) and with Dubovik's spheroid model (Dubovik et al., 2006) by assuming the spheroidal distribution of Gasteiger et al. (2011) for non-spherical particles (CNS). However, since HETEAC is optimized for the wavelength of EarthCARE's ATLID (355 nm) and the calculated properties hold for idealized spheres and spheroids, the scattering properties used in HETEAC-Flex have been slightly adjusted to meet the experimental findings (Floutsi et al., 2023). In particular, adjustments were made for the 532 nm backscatter coefficient per unit particle volume for the CNS aerosol component (a decrease of 0.013 in the case of CNS based on Saharan dust observations). In addition, the particle linear depolarization ratio for the FSA, FSNA and CS aerosol components, which is zero in HETEAC, was adjusted for HETEAC-Flex to better reflect the natural shape variability as found from the ground-based lidar observations. Furthermore, in HETEAC-Flex the CNS component is differentiated into Saharan, and Asian, which refers to coarse non-spherical particles originating from one of the aforementioned desert regions. As shown in Floutsi et al. (2023), dust originating from Central Asia exhibits significantly lower lidar ratios compared to Saharan dust due to differences in the source region mineralogy, i.e., varying content of iron oxide minerals, clay, etc. (e.g., Veselovskii et al., 2020). The particle linear depolarization ratio for the CNS aerosol component was also slightly decreased for 355 nm and increased for 532 nm in HETEAC-Flex. At the same time, the microphysical properties used in HETEAC-Flex (Table 1) were deliberately kept the same as in HETEAC, since the adjustments were small and consistency between these two approaches is important in terms of cross-validation of the algorithms and further support for the EarthCARE mission. As discussed in Wandinger et al. (2023a), experimental data cannot be fully described with the spheroidal shape model, which is the main reason for the necessary adjustments in the OEM application.



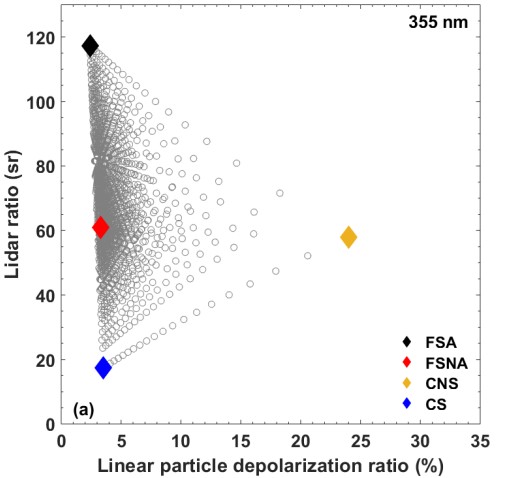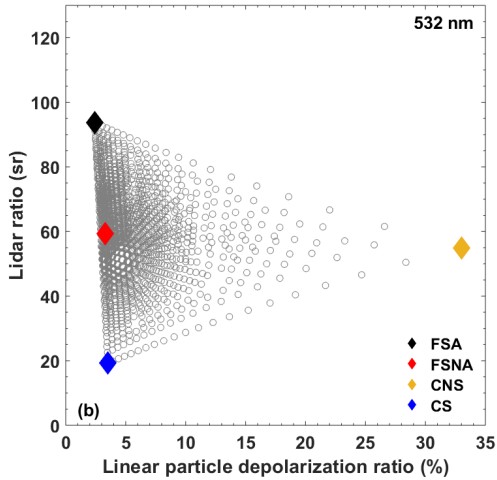

**Figure 2.** Simulated values of lidar ratio versus particle linear depolarization ratio at (a) 355 nm and (b) 532 nm for multimodal mixtures (grey open circles) of four basic aerosol components (rhombuses) based on Table 2. The depicted CNS component corresponds to Saharan dust.

Figure 2 shows two of the resulting intensive optical properties for the four basic aerosol components and their multimodal external mixtures obtained by using the optical properties of Table 2. The different aerosol mixtures have been calculated with a step width of 5 % in terms of relative volume (per aerosol component). The external mixing assumption leads to straight connecting lines in the optical parameter space. As it can be seen, most of the multimodal mixtures produce lidar ratios between 40 and 80 sr and linear depolarization ratios between 2.4 % and 5 %. Such values are indeed most often observed in nature (see

e.g., Fig. 2 and 3 in Floutsi et al., 2023). It should be noted that a considerable change of the optical parameters is observable only when one of the components starts dominating the mixture. In addition, a large contribution of dust is needed to cause a considerable particle linear depolarization ratio. Similarly, very large or very small lidar ratios are produced only when the small, spherical, strongly absorbing component or the coarse spherical component dominates, respectively. Similar behavior is observed for the intensive optical properties at 355 and 532 nm. However, it can be seen that there is a higher sensitivity of the

lidar ratio at 355 nm, while at 532 nm the linear depolarization ratio has higher discrimination power.

    Similarly to Fig. 2, Fig. 3 shows the extinction-related Ångström exponent versus the particle linear depolarization ratio (a) and lidar ratio (b), both at 355 nm, for the four basic aerosol components and their multimodal external aerosol mixtures. Again, the different aerosol mixtures have been calculated with a step width of 5 % in terms of relative volume (per aerosol component). The 2D spaces created by the different optical parameters are different compared to the respective ones in Fig. 2.

The coarse-mode aerosol components (i.e., CS and CNS) exhibit extinction-related Ångström exponent values of around zero, while the fine-mode aerosol components (i.e., FSA and FSNA) show extinction-related Ångström exponent values above unity. It can be seen that most multimodal external aerosol mixtures have extinction-related Ångström exponent values ranging





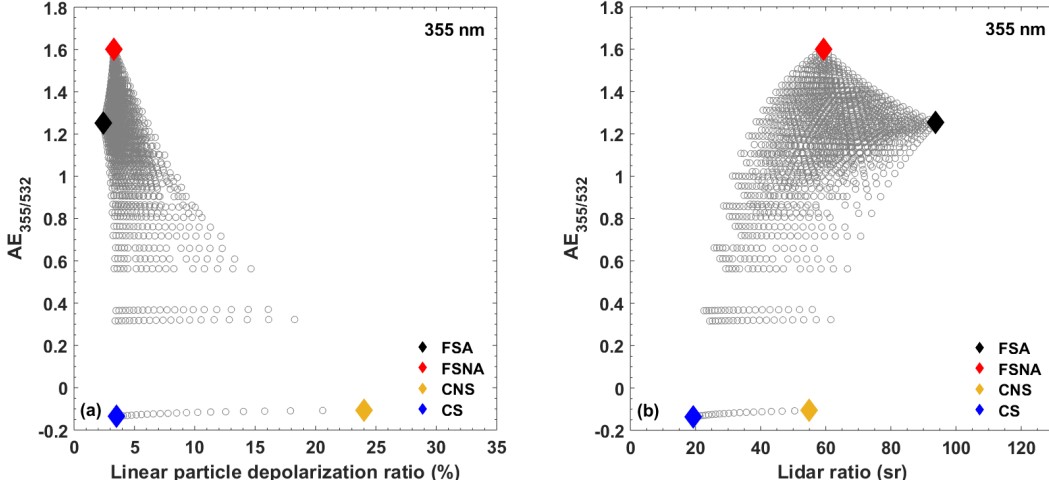

**Figure 3.** Same as Fig. 2, but for the extinction-related Ångström exponent versus the particle linear depolarization ratio (a) and lidar ratio (b) at 355 nm.

between 0.8 and 1.4. A small Ångström exponent requires high relative volume contributions of coarse-mode aerosol particles and therefore not so many aerosol mixtures (simulated) have low Ångström exponent values.

## 2.4 Forward model

To obtain the intensive optical parameters of multimodal aerosol compositions, mixing rules have to be applied. For each component, the scattering properties per unit particle volume are predefined (Table 2). The optical parameters of interest are then derived from the extinction and backscatter coefficients per unit volume ($\alpha_\lambda^*$, $\beta_\lambda^*$, respectively), the particle linear depolarization ratio ($\delta_\lambda$) of the aerosol components (indexed with $j$) and the relative volume contribution ($x$) of all components as follows (below, $\lambda$ has been omitted from the first two equations for the sake of clarity, as the scattering properties are at the same wavelength):

$$\delta = \frac{\sum x_j \beta_j^* \frac{\delta_j}{1+\delta_j}}{\sum x_j \beta_j^* \frac{1}{1+\delta_j}}, \text{ for the particle linear depolarization ratio} \tag{3}$$

$$S = \frac{\sum x_j \alpha_j^*}{\sum x_j \beta_j^*}, \text{ for the lidar ratio} \tag{4}$$

$$\mathring{A} = \frac{\ln\left(\frac{\sum x_j \alpha_{j,\lambda_1}^*}{\sum x_j \alpha_{j,\lambda_2}^*}\right)}{\ln\left(\frac{\lambda_1}{\lambda_2}\right)}, \text{ for the extinction-related Ångström exponent} \tag{5}$$





**Table 3.** The different available forward model configurations along with the required input parameters.

| Retrieval mode | Parameters |
|---|---|
| 1 | $\delta_{355}, S_{355}$ |
| 2 | $\delta_{532}, S_{532}$ |
| 3 | $\delta_{355}, S_{355}, \mathring{A}_{355/532}$ |
| 4 | $\delta_{532}, S_{532}, C_{\beta_{532/1064}}$ |
| 5 | $\delta_{355}, S_{355}, \delta_{532}, S_{532}$ |
| 6 | $\delta_{355}, S_{355}, \mathring{A}_{355/532}, \delta_{532}, S_{532}, C_{\beta_{532/1064}}$ |

$$C_{\beta_{\lambda_1/\lambda_2}} = \frac{\sum x_j \beta^*_{j,\lambda_1}}{\sum x_j \beta^*_{j,\lambda_2}} \text{ for the backscatter-related color ratio.} \tag{6}$$

All four equations presented above assemble the forward model and depending on the available measurements in the measurement vector (**y**), the forward model is adjusted accordingly (to simulate only the available measurements). Table 3 summarizes the six different predefined forward-model configurations or retrieval modes that are currently available. The different retrieval modes provide great flexibility in terms of available input and ensure a retrieval with a minimum amount of two pa-
rameters per measurement (both at the same wavelength). It should be noted that the nature of the algorithm is such that the forward operator can be easily modified and extended according to the user needs or the application considered. This feature is of great importance, especially for ground-based lidars, where the channel configuration might differ from instrument to instrument.

### 2.5 Optimum solution and convergence

The variational approach for obtaining an optimal estimate of the atmospheric state $\widehat{\boldsymbol{x}}$, given a measurement vector **y** and a forward model **F(x,b)**, is performed by minimizing a cost function of the form (Rodgers, 2000)

$$\boldsymbol{J}(\widehat{\boldsymbol{x}}) = \boldsymbol{J}_{\boldsymbol{\alpha}}(\widehat{\boldsymbol{x}}) + \boldsymbol{J}_{\boldsymbol{y}}(\widehat{\boldsymbol{x}}) + \boldsymbol{J}_{\mathrm{con}}(\widehat{\boldsymbol{x}}), \tag{7}$$

where $\boldsymbol{J}_{\boldsymbol{\alpha}}(\widehat{\boldsymbol{x}})$ represents the initial guess costs (or a priori costs), $\boldsymbol{J}_{\boldsymbol{y}}(\widehat{\boldsymbol{x}})$ the observation costs and $\boldsymbol{J}_{\mathrm{con}}(\widehat{\boldsymbol{x}})$ the penalty term to ensure physically meaningful retrievals of relative volume per aerosol component. While from a mathematical point of
view, relative volume contributions below 0 % or above 100 % are feasible, this is not the case from a physical point of view. Therefore, $\boldsymbol{J}_{\mathrm{con}}(\widehat{\boldsymbol{x}})$ adds a penalty, if the retrieval produces a relative volume per aerosol component that exceeds the interval 0 to 1, where the variables are continuous. The function is defined as

$$\boldsymbol{J}_{\mathrm{con}}(\widehat{\boldsymbol{x}}) = \begin{cases} 0 & \text{for } 0 \leq \boldsymbol{x}_{\mathrm{j}} \leq 1 \\ \\ \zeta |(\boldsymbol{x}_{\mathrm{j}})|^3 & \text{else,} \end{cases} \tag{8}$$



where $\boldsymbol{x}_j$ are the elements of the state vector (relative volume contributions of the different aerosol components) and $\zeta$ is a constant that is proportional to the strictness of the constraint. Here, $\zeta$ is set to a value large enough (i.e., $10^6$) to avoid relative volumes exceeding the interval boundaries. In addition to the penalty terms described above, if the retrieved relative volume of an aerosol type is smaller than 0, then it is automatically set to 0, and if the total relative volume contribution (sum of the relative volume contribution per aerosol component) is greater than 1, then the state vector is normalized, i.e., each value of the state vector $\boldsymbol{x}_j$ is divided by the sum of the relative volume of the four different aerosol components. There is no constraint in place in the case of a total relative volume contribution that is less than 1. In such cases, the remaining contribution is characterized as uncategorized aerosol. Most usually, when a total relative volume contribution less than 1 appears in the optimal-solution state, it is a result of the normalization of the state vector (in the previous step).

Expanding Eq. (7) we get:

$$\boldsymbol{J}(\widehat{\boldsymbol{x}}) = [\boldsymbol{x} - \boldsymbol{x}_\alpha]^T \boldsymbol{S}_\alpha^{-1}[\boldsymbol{x} - \boldsymbol{x}_\alpha] + [\boldsymbol{y} - \boldsymbol{F}(\widehat{\boldsymbol{x}})]^T \boldsymbol{S}_\epsilon^{-1}[\boldsymbol{y} - \boldsymbol{F}(\widehat{\boldsymbol{x}})] + \boldsymbol{J}_{\mathrm{con}}(\widehat{\boldsymbol{x}}). \tag{9}$$

The optimum solution can be found iteratively using the Levenberg–Marquardt method (LM), which is a combination of the gradient/steepest descent and Gauss–Newton methods:

$$\boldsymbol{x}_{i+1} = \boldsymbol{x}_i + [(1 + \gamma_i)\boldsymbol{S}_\alpha^{-1} + \boldsymbol{K}_i^T \boldsymbol{S}_\epsilon \boldsymbol{K}_i + \ddot{\boldsymbol{J}}_{\mathrm{con}}]^{-1} \times \{\boldsymbol{K}_i^T \boldsymbol{S}_\epsilon[\boldsymbol{y} - \boldsymbol{F}(\boldsymbol{x}_i)] - \boldsymbol{S}_\alpha^{-1}[\boldsymbol{x}_i - \boldsymbol{x}_\alpha] + \dot{\boldsymbol{J}}_{\mathrm{con}}\}, \tag{10}$$

with $i$ being the iteration step and the dots over $\boldsymbol{J}_{\mathrm{con}}(\widehat{\boldsymbol{x}})$ denoting the first and second derivative with respect to the state vector. $\boldsymbol{K}_i$ is the weighting function matrix, or Kernel or Jacobian (from now on Jacobian), defined as $\boldsymbol{K}_i = \frac{\partial \boldsymbol{F}(\widehat{\boldsymbol{x}})}{\partial \widehat{\boldsymbol{x}}}$ and calculated analytically for the lidar ratio and the linear particle depolarization ratio and numerically for the remaining quantities of the forward model, by perturbing the corresponding variable of the state vector by $10^{-3}$. The LM parameter ($\gamma$) is a factor that minimizes the cost function (Eq. (9)). When $\gamma \rightarrow 0$, the solution tends to the Gauss–Newton solution ($\gamma = 0$), while when $\gamma \rightarrow \infty$, the solution tends to the steepest-descent solution, allowing thus the solution to leave a local minimum towards a global minimum. In this study, the initial value for the $\gamma$ parameter is 2. It is increased by a factor of 10, if the cost function in the current iteration step is greater than the one in the previous step ($\boldsymbol{J}(\boldsymbol{x}_{i+1}) \geq \boldsymbol{J}(\boldsymbol{x}_i)$). It is reduced by a factor of 2, if the cost function is smaller ($\boldsymbol{J}(\boldsymbol{x}_{i+1}) < \boldsymbol{J}(\boldsymbol{x}_i)$). In retrospect, the LM method was found to converge not faster but more reliably than a Gauss–Newton iteration and, hence, it is preferred in this study.

The iteration procedure of Eq. (10) starts with the initial guess of the state vector ($\boldsymbol{x}_i = \boldsymbol{x}_\alpha$) and is repeated until the following criterion is fulfilled:

$$[\boldsymbol{F}(\boldsymbol{x}_{i+1}) - \boldsymbol{F}(\boldsymbol{x}_i)]^T \boldsymbol{S}_{\delta \widehat{y}}^{-1}[\boldsymbol{F}(\boldsymbol{x}_{i+1}) - \boldsymbol{F}(\boldsymbol{x}_i)] \ll d_f, \tag{11}$$

where $\boldsymbol{S}_{\delta \widehat{y}} = \boldsymbol{S}_\epsilon(\boldsymbol{K}\boldsymbol{S}_\alpha \boldsymbol{K}^T + \boldsymbol{S}_\epsilon)^{-1}\boldsymbol{S}_\epsilon$ is the covariance matrix between the measurement ($\boldsymbol{y}$) and $\boldsymbol{F}(\widehat{\boldsymbol{x}})$, and $d_f$ describes the degrees of freedom of the measurement, i.e., the number of independent observables (see. Table 3). In the algorithm, the "much smaller" mathematical operator ($\ll$) translates into $1/10$ of the degrees of freedom of the measurement. The degrees of freedom of the measurement are defined as $d_f = d_s + d_n$, where the first term is attributable to the state vector and the second term corresponds to the noise, and they can range between two and six degrees (depending on the input parameters).



Finally, the covariance matrix of the optimally estimated state vector (a posteriori) is calculated as follows:

$$\widehat{\boldsymbol{S}} = (\boldsymbol{K}^T \boldsymbol{S}_\epsilon \boldsymbol{K} + \boldsymbol{S}_\alpha^{-1})^{-1}. \tag{12}$$

The diagonal elements of $\widehat{\boldsymbol{S}}$ are the retrieval errors of the final optimal state vector $\widehat{\boldsymbol{x}}$. A pseudo-code summarizing the whole OEM procedure as described above is provided in Appendix B.

## 2.6 Statistical significance of the retrieved state

Once the iteration has converged, a Pearson's chi-squared ($\chi^2$) test must be carried out to ensure correct convergence (i.e., avoid convergence at a false minimum). This is done by comparing the forward-modeled measurements at the optimal state $\boldsymbol{F}(\widehat{\boldsymbol{x}})$ with the measurement vector $\boldsymbol{y}$, along with the corresponding error covariance matrix $\boldsymbol{S}_{\delta\widehat{y}}$:

$$\chi^2 = [\boldsymbol{F}(\widehat{\boldsymbol{x}}) - \boldsymbol{y}]^T \boldsymbol{S}_{\delta\widehat{y}}^{-1} [\boldsymbol{F}(\widehat{\boldsymbol{x}}) - \boldsymbol{y}]. \tag{13}$$

This statistical test is usually used for outlier identification (i.e., a retrieved state that does not belong to a Gaussian distribution). All retrievals for which the computed $\chi^2$ is greater than a threshold value $\chi_{thr}$ are discarded and all the rest, for which $\chi^2 \leq \chi_{thr}$, are accepted and further analysed. In this study, a significance level of 95% is selected and $\chi_{thr}$ is calculated for a 5% probability that $\chi^2$ is greater than the threshold for a theoretical $\chi^2$ distribution with $d_f$ degrees of freedom (Chi-Square Table, 2008). In other words, if the estimated retrieved state is found to be statistically significant within the 95% significance level, there is a 5% chance of not being true. However, the significance level can be adjusted easily according to the user needs.

## 3 Application of HETEAC-Flex

HETEAC-Flex is applied to three selected case studies to provide a more insightful overview of the algorithm's capabilities. The first case examined and presented below (Sect. 3.1) concerns a geometrically thick mineral dust layer observed during the A-LIFE (Absorbing aerosol layers in a changing climate: aging, LIFEtime and dynamics) field campaign, which took place at Cyprus in April 2017 (Weinzierl and the A-LIFE Science Team, 2021). Due to the characteristically high particle linear depolarization ratio, desert-dust-dominated aerosol layers are a fairly easy task for the typing scheme.

The second case regards two aerosol layers that were observed in January 2008 over Praia, Cabo Verde, during the SAMUM–2 field campaign (Sect. 3.2). The aerosol layers were characterized as a mixture of smoke and desert dust particles (Tesche et al., 2011a, b), and the corresponding aerosol contributions to the backscatter and extinction coefficients have been determined with the POLIPHON (polarization-lidar photometer networking) method (Tesche et al., 2009b; Mamouri and Ansmann, 2014). Therefore, this case study also facilitates a comparison between the POLIPHON and the HETEAC-Flex results to examine the consistency between the two methodologies.

The third case deals with three aerosol layers that were observed over Haifa, Israel in August 2018 (Sect. 3.3). The layers were stacked atop each other and were dominated by different aerosol types as indicated by the different optical properties they exhibited. In particular, the planetary boundary layer (PBL) was influenced by anthropogenic pollution, while the layer right



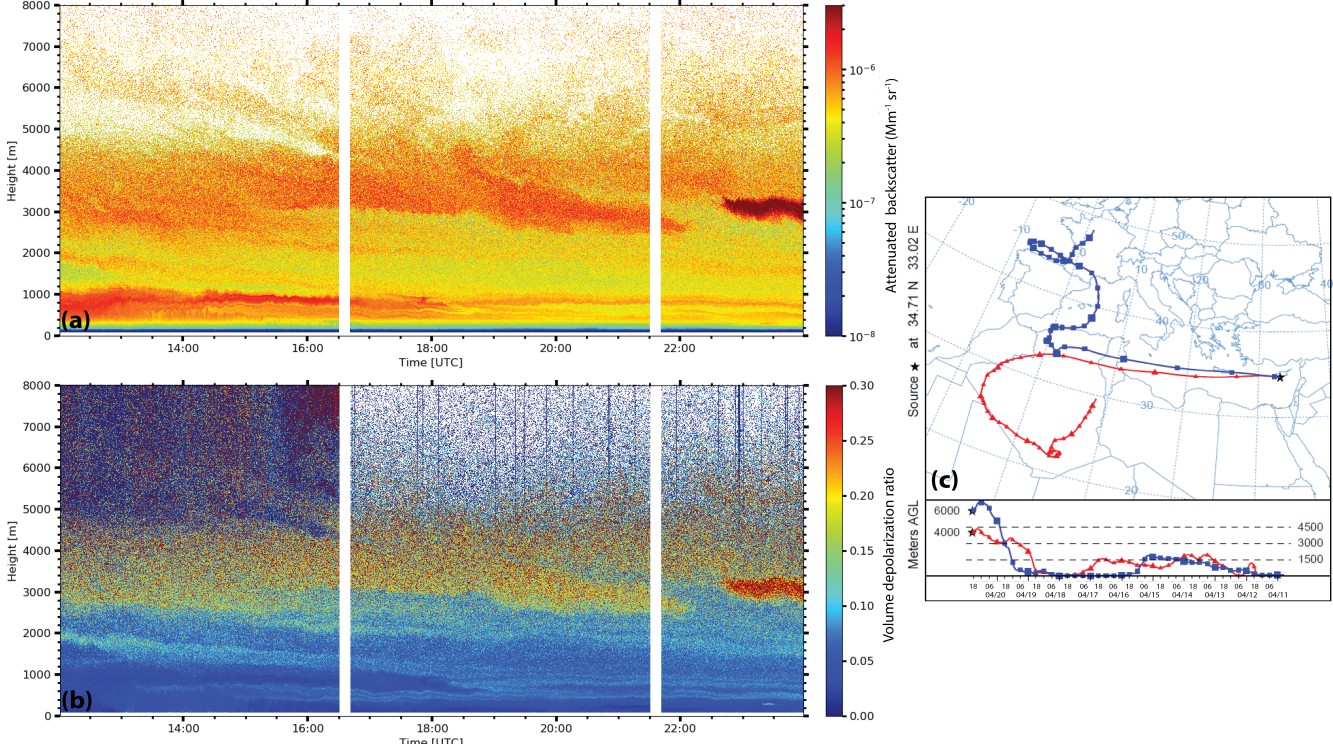

**Figure 4.** Overview of the atmospheric conditions, in terms of (a) range-corrected signal at 1064 nm and (b) linear volume depolarization ratio at 532 nm at Limassol, Cyprus on 20/04/2017, between 12:00 and 24:00 UTC. No data are available during regular depolarization calibration periods (white bars). (c) 10–day HYSPLIT backward trajectories ending at Limassol, Cyprus, on 20 April 2017, 19:00 UTC.

above the PBL was dominated by marine particles. The layer above was characterized as a lofted desert-dust layer. The optical properties of the layers as well as their characterization by HETEAC-Flex have been presented by Heese et al. (2021).

### 3.1 Case 1: Desert dust

#### 3.1.1 Overview

During the A-LIFE campaign, over a one-month measurement period (April 2017), several aerosol types were observed with a ground-based Raman lidar of type Polly$^{XT}$ (Engelmann et al., 2016; Baars et al., 2016), as part of LACROS (Leipzig Aerosol and Cloud Remote Observations System). LACROS was located in Limassol, Cyprus and operated on the premises of the Cyprus University of Technology (CyCARE campaign, Ansmann et al., 2019; Radenz et al., 2021a). Here, the focus is given to a temporally stable, approximately 8 km geometrically thick aerosol plume that was observed on 20 April 2017. Figure 4
shows the temporal development of (a) the range-corrected signal (at 1064 nm) and (b) the volume depolarization ratio (at 532 nm) on 20 April 2017, between 12:00 and 24:00 UTC. Increased backscattering is evident in parts of the aerosol layer.



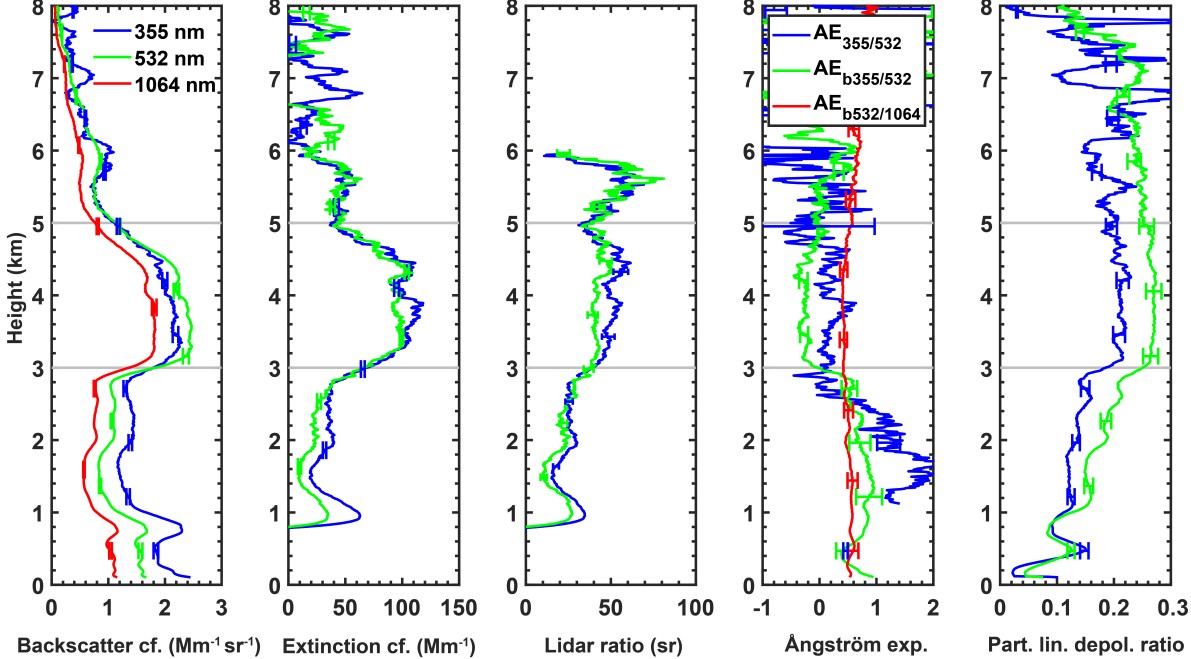

**Figure 5.** Vertical profiles of the particle backscatter and extinction coefficients, particle lidar ratio, Ångström exponents and particle depolarization ratio measured at Limassol, Cyprus, on 20 April 2017, from 17:00-19:00 UTC. A smoothing length of 750 m was used for the extinction and lidar ratio, and 200 m for the backscatter and depolarization ratio. Faint grey lines indicate the aerosol layer boundaries.

The aerosol layer observed above the lidar site originated from the Sahara region as indicated by the HYSPLIT (Stein et al., 2015) backward trajectories (Fig. 4c) arriving at 4 and 6 km. Air masses arriving at 4 km originated directly from Algeria and crossed Tunisia and the Mediterranean Sea on the way to Limassol. These air masses carried mainly desert dust particles. Air

masses arriving at higher altitudes (6 km) originated from Western Europe. However, four days prior to their observation at Limassol, the same air masses were located above Algeria at very low altitudes and, thus, could pick up desert dust particles (aerodynamic lifting).

The vertically-resolved lidar-derived optical parameters (between 17:00 and 19:00 UTC) are presented in Fig. 5. Maximum extinction coefficient values of approx. 120 Mm$^{-1}$ are observed in the core of the plume, between 3 and 5 km. The extinction

and backscatter-related Ångström exponents for the same altitudes were around 0.1 and 0, both for the wavelength pair of 355/532 nm, respectively. The lidar ratio was stable within the core (3-5 km) of the aerosol layer with values of approx. 49 and 44 sr for 355 and 532 nm, respectively. The particle linear depolarization ratio was high, exceeding 20 % at 355 and 532 nm above 3 km, indicating thus a strong presence of coarse-mode dust particles.



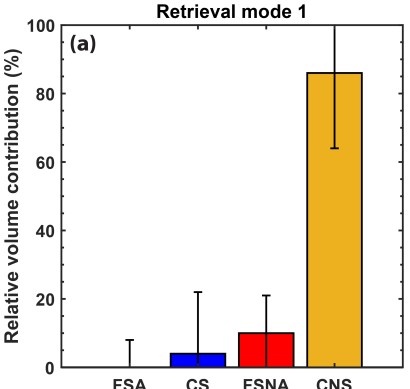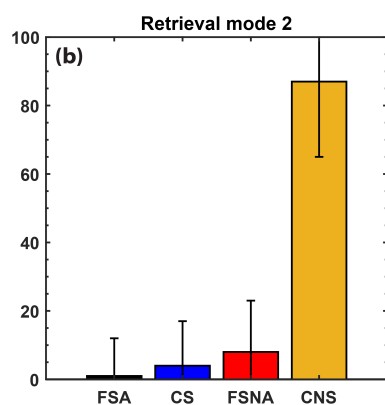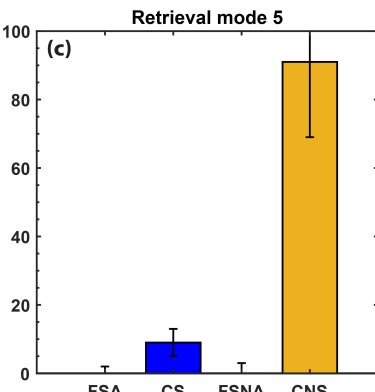

**Figure 6.** Mixing state of the four aerosol components as determined by HETEAC-Flex for the layer observed between 3 and 5 km at Limassol, Cyprus, on 20 April 2017, from 17:00-19:00 UTC for retrieval mode (a) 1, (b) 2 and (c) 5. It should be noted that the error bars have been confined between the constraint-accepted limits.

### 3.1.2 Aerosol characterization by HETEAC-Flex

Layer-averaged values (from 3–5 km) of lidar ratio ($49 \pm 8$ sr) and particle linear depolarization ratio ($20.6 \pm 2$ %) at 355 nm were used as input to HETEAC-Flex, and the retrieval mode based only on UV information (mode 1, see Table 3) was carried out. The optimal solution (visualized in Fig. 6a), which was also statistically significant within the 95 % confidence interval, confirmed the dominance of the CNS aerosol component in the aerosol mixture: in terms of relative volume contribution, the CNS component occupied $86 \pm 22$ % of the total aerosol mixture volume. The contributions of the remaining aerosol

components were small and almost negligible: $10 \pm 11$ % of FSNA, $4 \pm 18$ % of CS, and $0 \pm 8$ % of FSA aerosol particles. Overall, coarse-mode particles dominated the aerosol mixture with a total relative volume contribution of 90 %. It should be noted that the uncertainties hold only for the interval for which the relative volume contribution remains positive. For instance, the relative volume contribution of the FSNA component can range between 21 % and 0 %.

As more optical information was available than in the UV only, the determination of the aerosol mixing state was also

attempted based on retrieval modes 2 and 5 (see Table 3). The optimal solution for retrieval mode 2 (information on 532 nm only) was statistically significant, while the one for retrieval mode 5 (simultaneous 355 and 532 nm retrieval mode) was not statistically significant (Fig. 6b and c, respectively). The statistically non-significant solution occurred due to the inability of the retrieved aerosol mixing ratio to reproduce the particle linear depolarization ratio at 532 nm within the measurement error range. Simulating optical properties for non-spherical particles still remains a challenging task, even though there have been

many attempts to realistically model the shape of non-spherical particles (see discussion in Sect. 6.2.2 of Wandinger et al., 2023a). Regardless of the statistical significance of the solution, both retrievals captured successfully the predominance of the CNS aerosol component ($87 \pm 22$ % and $91 \pm 22$ % for retrieval modes 2 and 5, respectively).





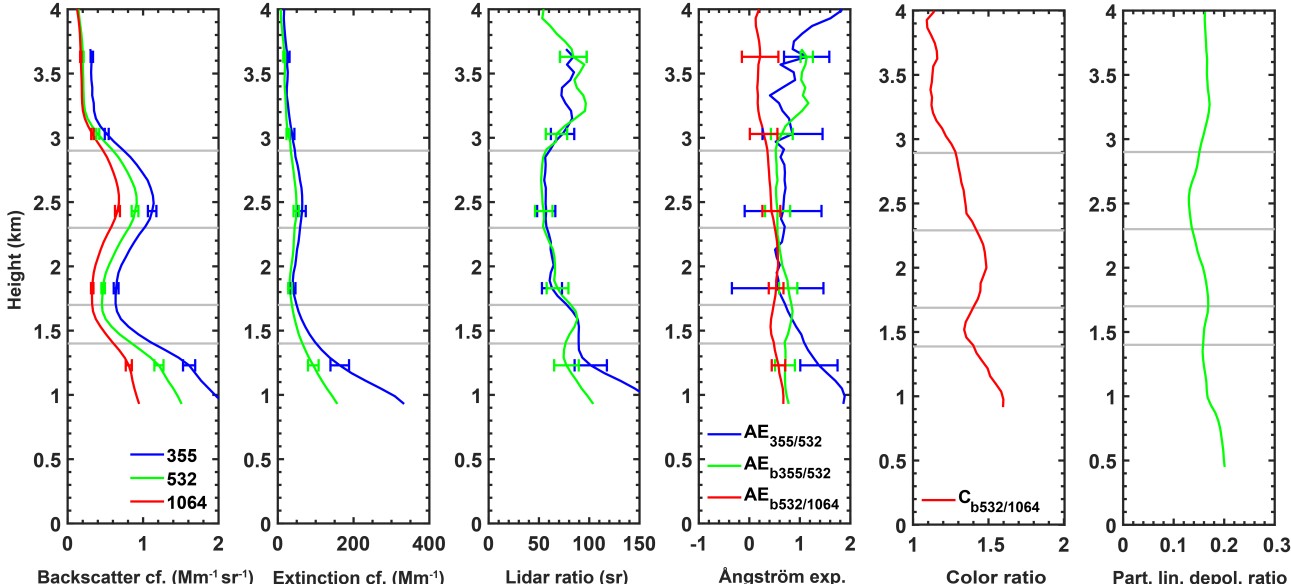

**Figure 7.** Vertical profiles of the particle backscatter and extinction coefficients, particle lidar ratio, Ångström exponents, backscatter-related color ratio and particle linear depolarization ratio measured at Praia, Cabo Verde, on 22 January 2008, between 20:20 and 22:30 UTC. A smoothing length of 660 m has been used (adapted profiles; analyzed by Tesche et al. (2011a, b)). Faint grey lines indicate the aerosol layer boundaries.

## 3.2 Case 2: Smoke and desert dust

### 3.2.1 Overview

During the SAMUM–2 campaign, a persistent and complex aerosol layering was observed above Praia, Cabo Verde, for almost one month (18 January to 14 February 2008), with the exception of a few days, where clean marine conditions dominated. The vast majority of the aerosol mixtures were observed above the PBL and consisted of smoke and dust particles, while pure layers were mostly confined in the lowest 1 km, and they were mostly pure dust layers. Here, the focus is put on a lofted layer, which was observed at altitudes between 1 and 3 km, on 22 January 2008, from 20:05–21:00 UTC. The same lofted layer has

been analyzed in detail by Tesche et al. (2011a).

On 22 January 2008, several aerosol layers were observed at altitudes up to 4 km (not shown here, see Fig. 6 of Tesche et al., 2011a). High volume linear depolarization ratio values in the lowest 0.8 km indicate the presence of desert dust particles (originating from northern Africa). Between 0.8 and 1 km, a decrease of the depolarization ratio is observed. Above 1 km altitude, the lofted layer has lower depolarization ratio values (compared to the lowest layer in altitude), and it has been

characterized as a mixture of mineral dust and biomass-burning/smoke aerosol originating from southern West Africa (Tesche et al., 2009b, 2011a, b).



**Table 4.** Contribution (in terms of relative volume) and respective uncertainties of the four basic aerosol components to the aerosol mixtures observed between 1.4–1.7 km (layer 1) and 2.3–2.9 km (layer 2) at Praia, Cabo Verde, on 22 January 2008, from 20:05–21:00 UTC. The retrieval mode is also indicated.

| Aerosol layer | Retrieval mode | HETEAC-Flex output (%) | | | |
|:---:|:---:|:---:|:---:|:---:|:---:|
| | | FSA | CS | FSNA | CNS |
| 1 | 2 | $25.8 \pm 15.4$ | $0 \pm 14.8$ | $0 \pm 17.6$ | $67.3 \pm 21.4$ |
| 1 | 4 | $28.5 \pm 16.0$ | $0 \pm 14.9$ | $0 \pm 17.2$ | $67.8 \pm 21.3$ |
| 2 | 2 | $1.7 \pm 11.7$ | $6.3 \pm 14.3$ | $14.3 \pm 17.7$ | $77.7 \pm 22.0$ |
| 2 | 4 | $1.7 \pm 12.1$ | $6.6 \pm 14.8$ | $17.8 \pm 17.6$ | $73.9 \pm 21.9$ |

Profiles of the backscatter and extinction coefficients, the lidar ratio, the Ångström exponent and the particle linear depolarization ratio are presented in Fig. 7 for a 2-h interval (20:20–22:30 UTC). Between 1 and 3 km altitude, two aerosol layers were identified. The aerosol layer boundaries have been defined based on the retrieved optical properties, as 1.4–1.7 km for the lower aerosol layer and 2.3–2.9 km for the upper aerosol layer. The averaged lidar ratios for the lower aerosol layer are $85.6 \pm 13.5$ sr and $84.2 \pm 13.3$ sr at 355 and 532 nm, respectively. The mean particle linear depolarization ratio at 532 nm for that layer is 0.16. The backscatter-related Ångström exponents at the wavelength pairs of 355/532 and 532/1064 nm range between 0.5 and 1. The optical properties of this aerosol layer are indicative of an aerosol mixture that is rather absorbing and contains both spherical and non-spherical particles. The upper layer exhibits mean lidar ratios of $57.0 \pm 9.0$ sr and $53.9 \pm 8.5$ sr at 355 and 532 nm, respectively. The layer-averaged particle linear depolarization ratio (532 nm) is 0.14. Both the extinction- and backscatter-related Ångström exponents range between 0.5 and 1, similar to the ones observed for the lower layer. The optical properties of the upper aerosol layer indicate again an aerosol mixture of spherical and non-spherical particles that is less absorbing compared to the lower aerosol layer. However, based solely on the aerosol measurements, the mixing ratio of the different aerosols contributing to the aerosol mixture observed cannot be determined.

### 3.2.2 Aerosol characterization by HETEAC-Flex

To characterize the mixing ratio (in terms of relative volume) of the different aerosol components contributing to the observed aerosol mixtures, HETEAC-Flex was applied to the two aforementioned aerosol layers (Fig. 7). Due to the absence of particle linear depolarization ratio information at 355 nm, the only possible retrieval modes were modes 2 and 4 (see Table 3). In addition to the aforementioned optical properties at 532 nm, backscatter-related color ratios of $1.4 \pm 0.5$ and $1.3 \pm 0.5$ (for layer 1 and layer 2, respectively, both at the wavelength pair 532/1064 nm) were used as inputs to the OEM-based typing scheme. It should be noted that the errors of the particle linear depolarization ratio (0.05) and the backscatter-related color ratio (0.5) had to be assumed (due to the absence of data) to enable the OEM retrievals.

The outcome of the different retrieval modes for the two aerosol layers is presented in Table 4. Both retrieval modes produced results that were statistically significant (within the 95 % confidence interval) for both aerosol layers and are therefore



considered for further analysis. It can be seen that in both aerosol layers, regardless of the retrieval mode, the CNS aerosol
component dominated in the aerosol mixture, in terms of relative volume, with values ranging from approx. 67 % to 78 %. The
second most abundant aerosol component was the FSA one ($\approx 26$ %–29 %) for the lower aerosol layer (layer 1) and the FSNA
one ($\approx 14$ %–18 %) for the upper aerosol layer (layer 2). The aerosol components least present in the lower aerosol layer were
the CS and FSNA ones, taking into account the retrieval uncertainties. For the upper aerosol layer, the least present aerosol
components were the CS and FSA ones. The results (Table 4) for both aerosol layers are consistent for both retrieval modes
(2 and 4). In addition, the results are consistent with the conclusions drawn by Tesche et al. (2011a). There, these layers were
attributed to a mixture of smoke (absorbing - FSA) and mineral dust (CNS) from West Africa.

For the lower aerosol layer (layer 1), both retrieval modes identified the aerosol mixture as a mixture of only two aerosol
components, CNS and FSA. Retrieval mode 2, which takes into account only the lidar ratio and particle linear depolarization
ratio at 532 nm, identified a CNS contribution of $67.3 \pm 21.4$ % and a FSA contribution of $25.8 \pm 15.4$ % (both in terms of
relative volume). Overall, the relative volume contribution of coarse-mode particles was 2.5 times higher than the one of the
fine-mode particles. While this solution is the statistically most-likely solution to produce the given measurements, the aerosol
components' contributions add up to a relative volume equal to 93.1 %. Therefore, the remaining 6.9 % relative volume can be
attributed as uncategorized aerosol. In addition to the optical parameters considered in retrieval mode 2, retrieval mode 4 takes
also into account the backscatter-related color ratio at the wavelength pair 532/1064 nm. The relative volume contribution of
the CNS aerosol component was $67.8 \pm 21.3$ % for retrieval mode 4 (almost identical with retrieval mode 2) and $28.5 \pm 16$ %
for the FSA aerosol component (slightly increased compared to retrieval mode 2). The mixing ratio of coarse- and fine-mode
particles slightly decreased compared to retrieval mode 2. The relative volume of the uncategorized aerosol was 3.7 % in
retrieval mode 4 (lower compared to retrieval mode 2). Given that both retrieval modes produced statistically significant results
for this layer and that retrieval mode 4 characterized a lower percentage of aerosol as uncategorized compared to retrieval mode
2, we can conclude that retrieval mode 4, which includes more optical parameters compared to retrieval mode 2, is preferred. In
general, when the quality of the input data is good, the retrieval mode that uses the most optical parameters is to be preferred,
as it provides more information to solve the ill-posed problem.

The aerosol mixture observed in the upper layer (layer 2) was identified by retrieval mode 2 as a mixture of primarily CNS
and FSNA aerosol with relative volume contributions of $77.7 \pm 22$ % and $14.3 \pm 17.7$ %, respectively. CS and FSA particles
also contributed to the aerosol mixture with much smaller relative volume contributions ($6.3 \pm 14.3$ % and $1.7 \pm 11.7$ %,
respectively). The relative volume contribution of the coarse-mode particles was 84 %, and it was 16 % for the fine-mode
particles. For the same aerosol layer, the results of retrieval mode 4 are similar to those of retrieval mode 2. The dominant
aerosol component was the CNS ($73.9 \pm 21.9$ %), followed by the FSNA ($17.8 \pm 17.6$ %). CS and FSA contributions to the
aerosol mixture were small ($6.6 \pm 14.8$ % and $1.7 \pm 12.1$ %, respectively). The mixing ratio of coarse- to fine-mode particles
was approx. 4.

To summarize the HETEAC-Flex results, both aerosol layers were identified by both retrieval modes as aerosol mixtures,
where coarse aerosol particles dominate in terms of relative contribution. The lower aerosol layer appears to be a mixture of
CNS and FSA aerosol particles with a mixing ratio of approx. 1.8. The upper layer appears to be a mixture of CNS and FSNA





**Table 5.** Contributing fractions of the different aerosol components to the particle OEM-estimated backscatter and extinction coefficient at 532 nm for the two aerosol layers observed at Praia, Cabo Verde, on 22 January 2008, from 20:20–22:30 UTC.

| Aerosol layer | Retrieval mode | Backscatter coef. 532 nm (%) | | | | Extinction coef. 532 nm (%) | | | |
|---|---|---|---|---|---|---|---|---|---|
| | | FSA | CS | FSNA | CNS | FSA | CS | FSNA | CNS |
| 1 | 2 | 59.8 | 0 | 0 | 40.2 | 71.7 | 0 | 0 | 28.3 |
| 1 | 4 | 62.1 | 0 | 0 | 37.9 | 73.6 | 0 | 0 | 26.4 |
| 2 | 2 | 3.9 | 10.2 | 40.2 | 45.7 | 6.7 | 3.6 | 43.6 | 46.1 |
| 2 | 4 | 3.7 | 9.8 | 46.3 | 40.2 | 6.3 | 3.4 | 50.1 | 40.2 |

aerosol particles with a mixing ratio of approx. 4. For this specific case, the consistency between the results obtained with the two retrieval modes (2 and 4) implies that the additional optical information (backscatter-related color ratio in the case of retrieval mode 4) did not significantly change the outcome, but also did not obstruct the analysis. In the following section, the HETEAC-Flex results for the two aerosol layers are compared with the results of the POLIPHON method to investigate the consistency between the two methodologies.

### 3.2.3 Comparison with POLIPHON

Following the POLIPHON methodology (Tesche et al., 2009b; Mamouri and Ansmann, 2014, 2016, 2017), the dust and smoke contributions to the measured total backscatter and extinction coefficient at 532 nm for the lofted aerosol layers observed on 22 January 2008 were calculated and presented by Tesche et al. (2011b). To compare the POLIPHON and HETEAC-Flex results, the OEM-based retrieval outputs had to be transformed into backscatter and extinction fractions. First, the extinction and

backscatter coefficient per aerosol component was calculated based on the a priori, component-specific backscatter coefficient. Then, the fraction attributable to the different components was computed with respect to the total OEM-retrieved backscatter and extinction coefficients. Table 5 summarizes the results of the aforementioned transformation. In addition, to facilitate the POLIPHON–HETEAC-Flex comparison, the POLIPHON-derived backscatter and extinction fraction vertical profiles had to be averaged for the two examined aerosol layers.

POLIPHON is able to separate only dust and non-dust components, which, in this case, are attributed to smoke particles. In HETEAC-Flex, the CNS aerosol component resembles desert dust particles and, therefore, can be directly compared with the dust fractions derived with the POLIPHON method. The non-dust fractions consist of contributions of the FSA, FSNA and CS aerosol particles. Both the FSA and FSNA aerosol components can resemble smoke particles with different absorption properties, such as in the case of smoke from different origins, e.g., from flaming or smoldering fires. The POLIPHON–

HETEAC-Flex comparison results are summarized in Fig. 8.

The OEM-derived results for the lower aerosol layer (layer 1, left panels in Fig. 8) agree to a satisfactory level with the POLIPHON results. POLIPHON results (stem with circle) indicate that the dust and smoke aerosol particles contributed almost



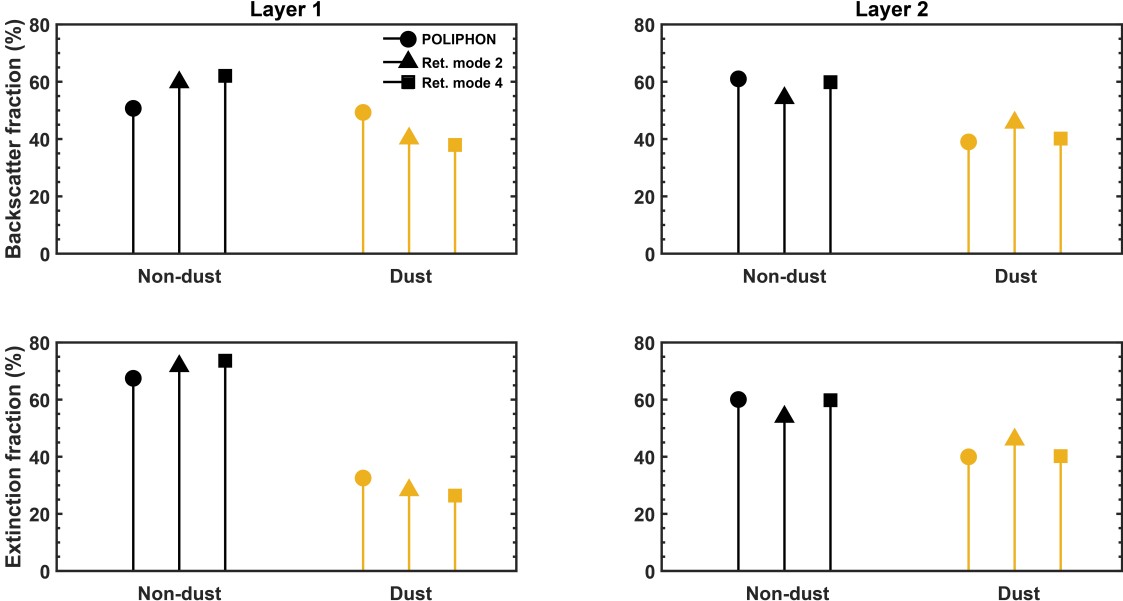

**Figure 8.** POLIPHON–HETEAC-Flex comparison of the fractions of backscatter (top panels) and extinction (bottom panels) coefficients for the lower (layer 1, left panels) and upper (layer 2, right panels) aerosol layers at Praia, Cabo Verde on 22 January 2008, between 20:20 and 22:30 UTC.

equally to the backscatter coefficient (50.7 % and 49.3 %, respectively), while the smoke particles dominated the extinction coefficient with contributions of 67.5 %. The HETEAC-Flex results are rather similar for both retrieval modes (2 and 4 indi-
cated by stems with triangle and square, respectively) and indicate that the FSA particles dominated both the backscatter and extinction coefficients with contributions of approx. 60 % and 72 %, respectively.

The POLIPHON results for the upper aerosol layer (layer 2, right panels in Fig. 8) indicate that the non-dust (smoke) particles dominated both the backscatter and extinction coefficients with contributions of 61 % and 60 %, respectively. The OEM-retrieved non-dust fractions of backscatter and extinction coefficients fit very well with the POLIPHON results. The non-
dust component clearly dominates both the backscatter and extinction coefficients for both retrieval modes. That is especially true for retrieval mode 4, where the results from both methodologies are almost identical. The non-dust particles dominate the backscatter (61 % for POLIPHON and 59.8 % for HETEAC-Flex) and extinction (60 % for POLIPHON and 59.8 % for HETEAC-Flex) coefficients.

Overall, the comparison between the POLIPHON and the HETEAC-Flex results is satisfactory, as the results compared very
well to each other (see the upper aerosol layer, retrieval mode 4 in Fig. 8). Overall, the advantage of HETEAC-Flex compared to POLIPHON is that HETEAC-Flex provides information on the radiative properties of the aerosol particles (i.e., whether the particles are absorbing or not). For instance, regarding the upper aerosol layer, POLIPHON assigned the entire amount of non-dust particles to smoke, while HETEAC-Flex provided the most-likely contributions of the remaining three components.



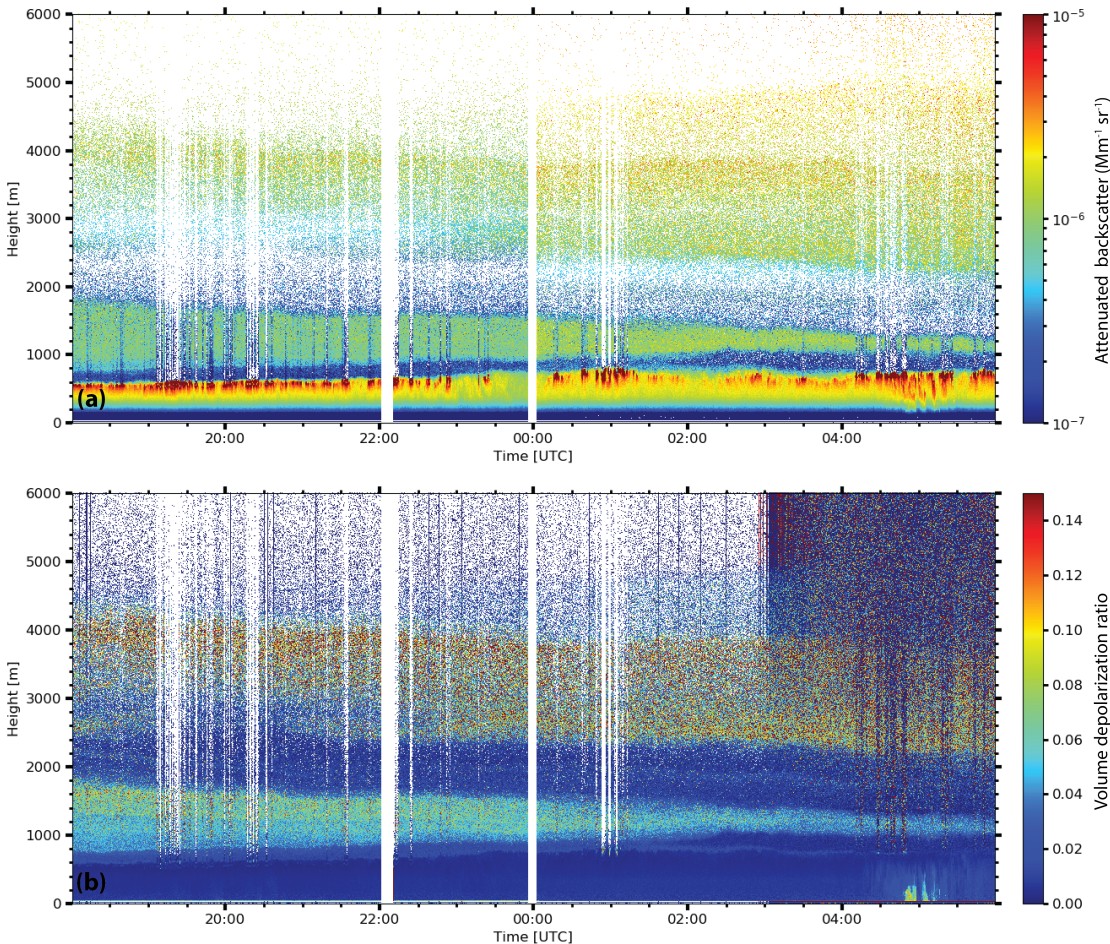

**Figure 9.** Range-corrected signal at 1064 nm (a) and volume linear depolarization ratio at 532 nm (b) above Haifa on 30 August 2018 at 18:00 UTC to 31 August 2018 at 06:00 UTC.

For the given optical properties, HETEAC-Flex suggested that the non-dust particles in the upper aerosol layer (see Table 4) are
more likely to be less absorbing (FSNA) than strongly absorbing (FSA), which provides additional insight to the composition of the smoke plumes observed over Praia, Cabo Verde.

### 3.3   Case 3: Multiple aerosol layers

A Polly$^{XT}$ lidar, part of PollyNET (Baars et al., 2016), operated in Haifa, Israel for two years continuously Heese et al. (2021). During the night of 30 to 31 August 2021, three aerosol layers were observed stacked atop each other above Haifa, Israel
(Fig. 9). The first layer is the PBL and reaches up to 0.9 km. The PBL is characterized by a strong backscatter signal and, in combination with very low particle linear depolarization ratio values, indicates the presence of spherical particles. The second layer (layer 2) extends between 0.9 and 2 km (thins out throughout the night), has a strong backscatter signal and consists of



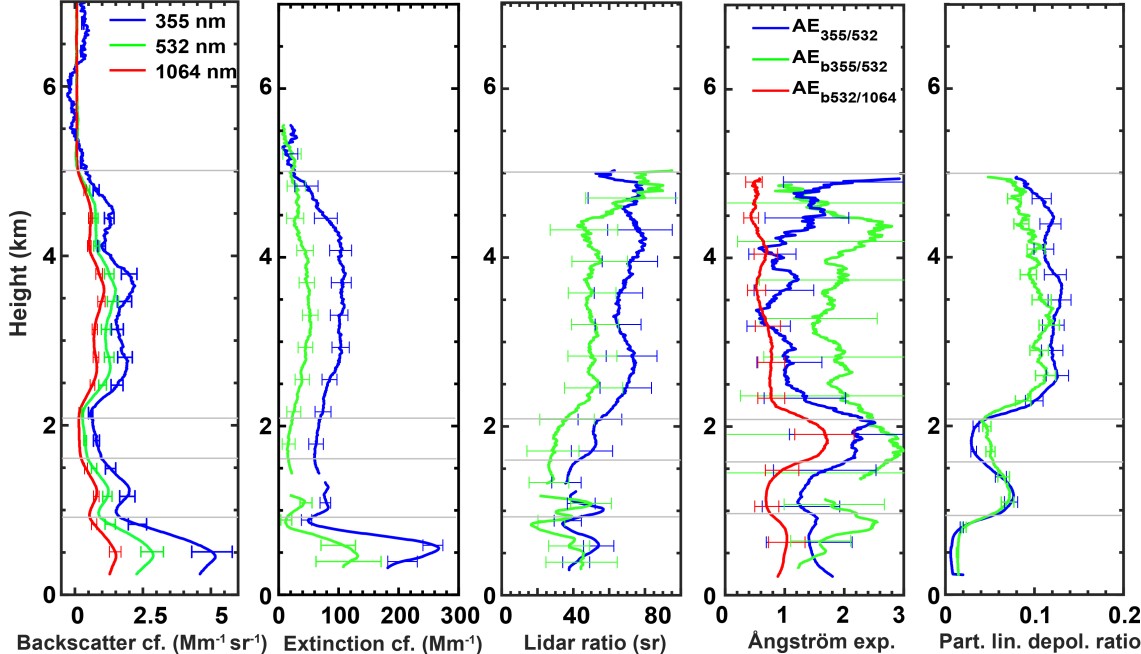

**Figure 10.** Vertical profiles of the particle backscatter and particle extinction coefficient, particle lidar ratio, Ångström exponents and particle linear depolarization ratio measured at Haifa, Israel, on 31 August 2018, from 01:20-02:44 UTC. A smoothing length of 382.5 m was applied and the indicated height range is starting above the measurement site.

slightly depolarizing particles. The last distinctive layer (layer 3) extends between 2.1 and 5 km, and apart from the strong backscatter signal it exhibits moderate depolarization values, indicating the presence of non-spherical particles in the layer.

Aerosol optical properties (Fig. 10) were derived between 01:20 and 02:44 UTC and have been presented and discussed in detail by Heese et al. (2021).

Based on the lidar ratio and the particle linear depolarization ratio at 532 nm (shown in Table 6), the aerosol components present in each layer were estimated by applying HETEAC-Flex (retrieval mode 2). The relative volume contribution of each aerosol component for the three aerosol layers is also shown in Table 6 (all results were statistically significant at 95 % confi-

dence interval) and visualized in Fig. 11. The most dominant aerosol component identified by HETEAC-Flex for the lowermost layer (PBL) was the FSNA aerosol component, with a relative volume contribution of $86 \pm 22$ %. The rest of the aerosol components contributed with very small relative volumes. The dominance of the FSNA aerosol component, which is associated with aerosol of anthropogenic origin, can be explained due to the neighboring of Haifa (to the east) with big industries (e.g., large petrochemical plants, cement factories, oil-fueled power station), smaller industries and workshops (Ganor et al., 1998).

In the weakly depolarizing aerosol mixture of the middle layer (layer 2), the most abundant aerosol component was the CS one. This component can be attributed either to sea-salt particles, expected to be found in abundance since Haifa is a coastal city, or to other aerosol types, e.g., anthropogenic pollution that has undergone hygroscopic growth. The second more abundant



**Table 6.** Aerosol optical properties (at 532 nm) and composition of the layers identified on 31 August 2018 (Heese et al., 2021).

| Aerosol layer | $S$ (sr) | $\delta$ | FSA | CS | FSNA | CNS |
|---|---|---|---|---|---|---|
| PBL | 40 | 0.01 | 2±9 % | 8±20 % | 86±22 % | 4±21 % |
| 2 | 30 | 0.07 | 12±13 % | 71±22 % | 8±20 % | 9±19 % |
| 3 | 50 | 0.12 | 1±12 % | 9±15 % | 16±17 % | 74±21 % |

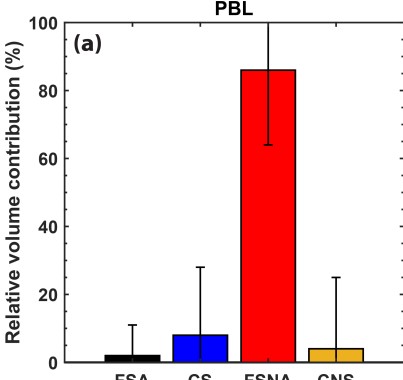
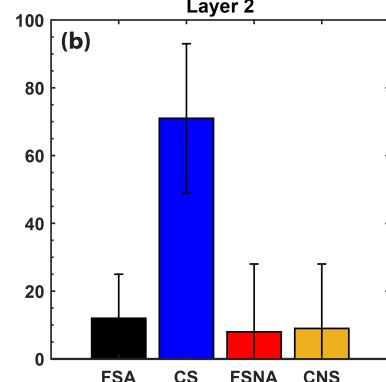
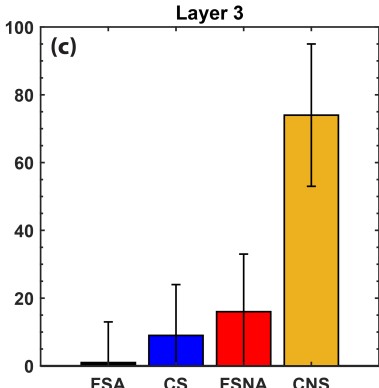

**Figure 11.** Mixing state of the four aerosol components as determined by HETEAC-Flex for the three aerosol layers observed at Haifa, Israel, on 31 August 2018, from 01:20-02:44 UTC. It should be noted that the error bars have been confined between the constraint-accepted limits.

aerosol was the one of fine mode (FSA and FSNA), originating from the aforementioned industrial sources. In the upper and geometrically thicker aerosol layer (layer 3), the most dominant aerosol component was the CNS with a relative volume contribution of $74 \pm 21$ %. The second most contributing component to the aerosol mixture was the FSNA one, followed by the CS aerosol component, suggesting that desert dust was mixed with particles of anthropogenic and marine origin. The temporally and vertically resolved air-mass source attribution TRACE (Radenz et al., 2021b) was also used by Heese et al. (2021) to identify the aerosol sources (not shown here). The results of TRACE support the HETEAC-Flex findings.

## 4 Conclusions and outlook

Within the scope of this work, HETEAC-Flex, a novel aerosol-typing scheme applicable to ground-based and spaceborne lidars, was developed based on the optimal estimation method. HETEAC-Flex enables the identification of up to four different aerosol components present in an aerosol mixture as well as the determination of their contribution to the aerosol mixture in terms of relative volume. The four aerosol components considered represent the most commonly observed aerosol particles in nature: fine-mode absorbing (FSA) and less absorbing (FSNA) particles, and coarse-mode spherical (CS) and non-spherical



(CNS) particles. The input parameters of HETEAC-Flex are lidar-derived intensive optical parameters, i.e., the lidar ratio and the particle linear depolarization ratio at two distinctive wavelengths (355 and 532 nm), the backscatter-related color ratio (for the wavelength pair of 532/1064 nm) and the extinction-related Ångström exponent (for the wavelength pair of 355/532 nm). The output of the algorithm, i.e., the relative volume contribution of each aerosol component, from a mathematical point of view is the most likely state that can reproduce the measurement vector, which contains the optical parameters measured by the

lidar. Once retrieved and statistically significant (in this study within the 95 % confidence interval), the (relative volume) contributions of the four different aerosol components can be used to estimate other quantities, such as the volume size distribution or the refractive index.

The performance and functionality of the algorithm were demonstrated through its application to three case studies of ground-based lidar observations. In the first case (Sect. 3.1), HETEAC-Flex was able to identify the dominance of the CNS

component in a lofted dust aerosol layer above Limassol, based on three different retrieval modes. In the second case (Sect. 3.2), in addition to the identification of the different aerosol components in the aerosol mixtures, HETEAC-Flex was compared to POLIPHON. While both methodologies are in good agreement, HETEAC-Flex can also separate the non-dust aerosol. In the third case (Sect. 3.3), HETEAC-Flex clearly identified the different compositions of the three aerosol layers (pollution-, marine- and dust-dominated) and, thus, gives additional value for aerosol research using lidar observations. Furthermore,

while not shown here, the algorithm has been applied to the complete 2-year-long lidar dataset from Haifa and it has led to the successful identification of the aerosol load above the region and its seasonal characterization (Heese et al., 2021).

The information content of the measurement vector (expressed by the degrees of freedom, see Sect. 2.5) was one of the driving forces behind the different available forward-model configurations (i.e., different combinations of intensive properties, see Table 3), which provide this typing scheme with great flexibility. In principle and while not explicitly shown here, each

retrieval mode is able to produce statistically significant results and there is no evidence suggesting that one mode is superior to another systematically. In retrospect, it can be concluded that the retrieval modes that include two input parameters (i.e., retrieval modes 1 and 2) tend to converge faster (but not necessarily more reliably) than those that include more input parameters. The conclusion is coherent since the more input parameters the more challenging the task of the forward model to simultaneously reproduce all the input parameters within the measurement error.

The retrieved contributions were sometimes accompanied by rather high values of the respective retrieval error (ranging between 8 % and 22 %). The a posteriori uncertainty (i.e., the covariance matrix of the optimally estimated state vector) is directly linked to the a priori uncertainty, meaning that the larger the a priori uncertainty the larger the retrieved a posteriori uncertainty. The retrieval is, therefore, strongly driven by the a priori uncertainty, which essentially constrains the retrieval solution space. Hence, the continuation of the high-quality lidar measurements with low uncertainties of the different aerosol

types as well as the expansion of the experimental collection introduced by Floutsi et al. (2023) is of great importance for this typing scheme.

The output of HETEAC-Flex can be used for several additional products, which are not presented nor demonstrated in this paper (e.g., the volume and number concentration per aerosol component). This choice was made deliberately as in this



paper the focus is given to the methodology behind the typing algorithm. However, a follow-up paper that focuses on the
intercomparison between the products obtained with HETEAC-Flex and in-situ measured quantities is planned.

*Code availability.* The HETEAC-Flex source code will be available via GitHub after the reviewing process is complete. For the lidar data visualization (Fig. 4 and 9), pyLARDA was used (https://doi.org/10.5281/zenodo.4721311, Bühl et al., 2021).

*Data availability.* The Polly$^{XT}$ lidar products are publicly available at https://polly.tropos.de/, last access: 17 August 2023.





**Appendix A: Decision tree**

The initial guess $\boldsymbol{x}_\alpha$ that is used to kick off the OEM is the output of a decision tree (Fig. A1), which is created based on the two intensive optical parameters with the highest discriminatory power - the particle linear depolarization ratio ($\delta$) and the lidar ratio $S$. The root node (topmost node) contains the mean $S$ and $\delta$ values (either at 355 or 532 nm) for the aerosol layer of interest. The first splitting parameter is the particle linear depolarization ratio ($\delta$), which has been already found to have the highest discriminatory power (e.g., Burton et al., 2012; Papagiannopoulos et al., 2018). The highest value of particle

linear depolarization ratio considered is 0.35, and aerosol particles exhibiting particle linear depolarization ratios higher than that, such as volcanic ash, are at the moment not considered in HETEAC-Flex and, therefore, are excluded from the decision tree. Further splitting is then done by the lidar ratio $S$. In the end, the terminal nodes contain the different labels along with the corresponding values of the initial guess of the state vector $\boldsymbol{x}_\alpha$. While the terminal node labels contain the names used for the aerosol components, they should not be confused with the components themselves and therefore are accompanied with an

asterisk (*). The asterisk indicates which aerosol component (or components in the case of mixtures) should be considered as dominant in the initial guess of the state vector $\boldsymbol{x}_\alpha$. For instance, the label CS* indicates that based on the lidar observations, an initial guess of the state vector where CS particles are dominant should be considered. The different sets of the initial guess of the state vector are provided in Table A1.

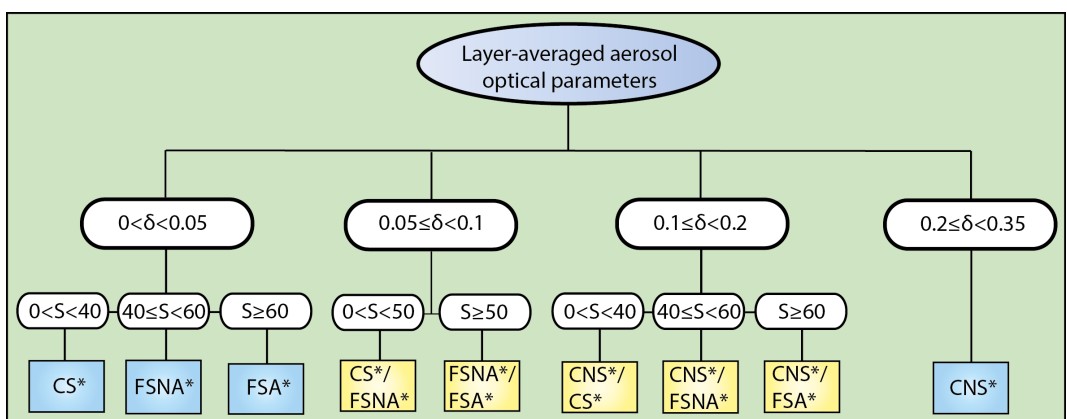

**Figure A1.** Decision tree used in HETEAC-Flex for the determination of the initial guess of the state vector ($\boldsymbol{x}_\alpha$). Unit for $S$ is sr.

      The relative volume contributions of each aerosol component used as the initial guess of the state vector $\boldsymbol{x}$ (Table A1) are

chosen to some extent arbitrarily (i.e., there is no way to determine the exact relative volume contributions), but at the same time they clearly reflect the dominant components in the state vector. As a general rule, when the decision tree returns a terminal node label with only one aerosol component as dominant, then its relative volume contribution is set to 0.85 and all the other components are set to 0.05, with the exception of the CNS component. When the CNS aerosol component is dominant, then its relative volume contribution is set to 1. Mixtures with two components are treated in a way that takes into account the particle

linear depolarization ratio values. Mixtures that do not include the CNS component (i.e., associated with depolarizing particles)





**Table A1.** Sets of initial guesses for the state vector ($\boldsymbol{x}_\alpha$), based on the terminal node label of the decision tree (Fig. A1).

| Terminal node label | FSA | CS | FSNA | CNS |
|---|---|---|---|---|
| CS* | 0.05 | 0.85 | 0.05 | 0.05 |
| FSNA* | 0.05 | 0.05 | 0.85 | 0.05 |
| FSA* | 0.85 | 0.05 | 0.05 | 0.05 |
| CS*/FSNA* | 0.0 | 0.5 | 0.5 | 0.0 |
| FSNA*/FSA* | 0.5 | 0.0 | 0.5 | 0.0 |
| CNS*/CS* | 0.0 | 0.7 | 0.0 | 0.3 |
| CNS*/FSNA* | 0.0 | 0.0 | 0.7 | 0.3 |
| CNS*/FSA* | 0.7 | 0.0 | 0.0 | 0.3 |
| CNS* | 0.0 | 0.0 | 0.0 | 1.0 |

have equal contributions of the two dominant components (relative volume contributions of 0.5 for each) and the other two components are set to 0. In mixtures where the CNS component appears as one of the dominant two components, the relative volume contributions of these components is distributed as 0.7 and 0.3, with the lowest contribution being assigned to the CNS component. That is because the CNS component is the only component with such high value of particle linear depolarization ratio and depending on the case, the OEM might need to adjust (through the corresponding Jacobian) quite a lot the initial guess before convergence is met. However, it should be noted that the sets of initial guesses are not unalterable, meaning that depending on the measurement vector the relative volume contributions in the terminal nodes can be easily adjusted.

**Appendix B: HETEAC-Flex pseudocode**

A step-by-step description of the algorithm follows. The program code is written in the proprietary programming language MATLAB (MATLAB, 2018).

1. Define the input measurements along with the corresponding error ($\boldsymbol{y}$ and $\boldsymbol{\epsilon}$, respectively).

2. The measurements are used as input for a decision tree that returns a first guess for the state vector ($\boldsymbol{x}_\alpha$).

3. For the given first guess of the state vector calculate via the forward model the measurements $\boldsymbol{F}(\boldsymbol{x}_\alpha)$, the Jacobian $\boldsymbol{K}$ and the covariance matrix between the measurement and the forward-modeled measurement $\boldsymbol{S}_{\delta\widehat{y}}$.

4. For a maximum of 30 iteration steps (start of the iterative process using the Levenberg–Marquardt method)

   (a) Initialize $\gamma$ parameter. Calculate the next state vector $\boldsymbol{x}_i$, new Jacobian $\boldsymbol{K}_i$, new modeled measurements $\boldsymbol{F}_i$ for the given $\boldsymbol{x}_i$, new covariance matrix between the measurement and the modeled measurement.

   (b) Calculate the cost function $\boldsymbol{J}(\boldsymbol{x}_i)$.



(c) If the cost function is greater than or equal to the one calculated in the previous step, increase the $\gamma$ parameter by a factor of 10. If it is smaller, decrease the $\gamma$ parameter by a factor of 2.

(d) Calculate the convergence criterion. Once its value is lower than $d_f/10$, stop the iteration.

5. Return the optimal state vector $\widehat{\boldsymbol{x}}$ along with the retrieval errors $\widehat{\boldsymbol{S}}$.

6. Perform $\chi^2$ test. Discard the results, if the retrieved state is not statistically significant within the desired significance level. Consider starting the iterative process again with a new $\boldsymbol{x}_\alpha$ or with different a priori settings.

*Author contributions.* AAF has developed the aerosol-typing scheme HETEAC-Flex and drafted the manuscript. UW has developed the HETEAC model, on which HETEAC-Flex is based. HB has provided guidance throughout the study. All authors contributed to the discussions during the development of HETEAC-Flex and to the paper.

*Competing interests.* Ulla Wandinger is a member of the editorial board of Atmospheric Measurement Techniques. The authors have no other competing interests to declare.

*Acknowledgements.* The authors gratefully acknowledge the NOAA Air Resources Laboratory (ARL) for the provision of the HYSPLIT transport and dispersion model used in this publication. The authors would like to thank Andreas Foth for his valuable input during various discussions on the topic of optimal estimation, and Moritz Haarig, Matthias Tesche and Birgit Heese for the data analysis from Limassol, Praia and Haifa, respectively.



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
