# Peer review of "HETEAC-Flex: An optimal estimation method for aerosol typing based on lidar-derived intensive optical properties"

_EGUsphere, 2023_

## Author Comment (AC1)

**Response to Referee #2**

**We thank the referee for their review, which helped us improve the manuscript. Below are the original comments in italics with our responses in bold text.**

*The manuscript presents the algorithm for analysis of aerosol mixture composition, based on expected EarthCare observations. Authors consider 4 main aerosol components (two types of fine and two types of coarse mode particles) and try to present the observed particle volume as a sum of these components volume, using OEM approach. Atmospheric aerosol is a complicated object and particle parameters are strongly variable, still this looks like a reasonable approach, because number of independent observations from EarthCare will be very limited. Manuscript is well written, contains new important results and is suitable for AMT. For me, personally, was interesting to see, how additional 532 nm measurements will influence the results obtained from UV channels.*

**Thank you for the positive feedback on our work.**

*Technical comments.*

*p.6. Choice of particle model parameters is always an issue. In particular, parameters depend on RH. This especially critical for maritime aerosol and for small non-absorbing particles. Probably authors could mention range of RH, where their model is applicable.*

**Indeed, the dependence of the microphysical properties on the relative humidity is not mentioned in the paper, because it is explored in detail in Wandinger et al., 2023a. For clarity, we have included the following sentence in the revised manuscript (lines 170-172): "The justification for the choice of microphysical parameters is described in Wandinger et al. (2023a), including a discussion on the relative humidity dependence of the size and refractive index of the particles."**

*Table.2. Authors assume that depolarization ratio of small particles is 3%. In practices, however, we often observe smoke and sulfate particles with depolarization up to ~8% or even higher. Small values of assumed depolarization ratio probably may increase contribution of dust particles in the examples provided in Fig.11.*

*For smoke particles the lidar ratio at 532 nm is usually significantly larger than at 355 nm. Can it be obtained from component parameters in Tab.2? The same question is for dust. In experiment lidar ratio of dust at 355 sometimes significantly higher than at 532.*

**In HETEAC, the model on which HETEAC-Flex is based, the particle linear depolarization ratio for all three spherical components is 0%. However, the scattering properties that were used in HETEAC-Flex were slightly adjusted to better reflect the experimental observations (Floutsi et al., 2023). These adjustments are also explicitly discussed in lines 169-175 of the original manuscript. Indeed, the particle linear depolarization ratio of the pure fine, spherical non-absorbing (FSNA) component is approx. 3% at 355 and 532 nm. This aerosol component resembles more pollution-related aerosol than smoke (the latter is absorbing and, thus, associated with the FSA component). Our long-term lidar-based collection of intensive aerosol properties indicates that the particle linear depolarization ratio of pollution-related aerosol is around 1.1 ± 0.3 and 2.8 ± 1 % for 355 and 532 nm, respectively (see Table 1 and Sect. 3.1.5 in Floutsi et al., 2023). Therefore, the choice of particle linear depolarization ratio is justified.**

A particle linear depolarization ratio of approx. 8% appears to be rather high for spherical particles. In the aforementioned case of smoke, it would indicate the presence of non-spherical soot particles in the aerosol mixture, usually observed in the upper troposphere (e.g., Burton et al., 2015).

However, in the case of elevated particle linear depolarization ratio, as correctly noticed, the contribution of the coarse, non-spherical (CNS) component will increase. One important aspect is that while the CNS component resembles dust particles, it does not necessarily imply the presence of dust. Other non-dust particles may also be coarse mode and non-spherical.

Based on the scattering properties of Table 2, the wavelength dependence of the lidar ratio (higher 532-nm/lower 355-nm) for smoke particles cannot be captured. This wavelength dependence is the most significant signature of aged smoke and stratospheric smoke, rather than fresh tropospheric smoke. In addition, the wavelength dependence would only impose a problem in the HETEAC-Flex retrievals for the retrieval modes that utilize the information from both wavelengths simultaneously, i.e., modes 5 and 6. With that being said, we should mention that we are aware of this issue and that currently there is ongoing research within the framework of the CARDINAL (Clouds, Aerosol, Radiation – Development of INtegrated Algorithms) project, aiming to define the microphysical/optical properties of stratospheric aerosol components (HETEAC-2) for EarthCARE.

The wavelength dependence of the lidar ratio for dust is well captured by HETEAC and can be resolved by HETEAC-Flex.

*Ln.231. As I understand, sometimes contribution of a single component may exceed 1.0. Just wonder, if it is possible to add additional constraint, that the sum of all components must be 1.0?*

In theory the contribution of a single component may exceed 1, but this is already tackled by a specific constraint, i.e., the constraint described by Eq. 8 prohibits that from happening. As explained in lines 231- 242 of the original manuscript, if the relative volume contribution of an aerosol component exceeds 1, then the penalty function adds a large term to the cost function, and therefore, for this solution the cost function is not minimal.

*Ln.238."… if the total relative volume contribution (sum of the relative volume contribution per aerosol component) is greater than 1, then the state vector is normalized"*

*This is actually related to my previous question.*

Indeed, it is related to the previous question, and is one of the constraints to prevent the issue you raised.

*Ln.239. "…There is no constraint in place in the case of a total relative volume contribution that is less than 1."*

*Why no normalization this case?*

As stated in lines 240-241 of the original manuscript version, in case of the total volume contribution of less than 1 we chose to attribute the missing volume contribution to uncategorized aerosol rather than normalizing the state vector. We chose to do so to avoid over-constraining the solution space.

*Fig.5. Probably no reason to show lidar ratio and extinction below 1 km.*

**Thank you, we have updated the Figure accordingly. In addition, due to a suggestion from Referee #3 now Fig. 5 includes the backscatter-related color ratio.**

**References**

Wandinger, U., Floutsi, A. A., Baars, H., Haarig, M., Ansmann, A., Hünerbein, A., Docter, N., Donovan, D., van Zadelhoff, G.-J., Mason, S., and Cole, J.: HETEAC – the Hybrid End-To-End Aerosol Classification model for EarthCARE, Atmos. Meas. Tech., 16, 2485–2510, https://doi.org/10.5194/amt-16-2485-2023, 2023a.

Floutsi, A. A., Baars, H., Engelmann, R., Althausen, D., Ansmann, A., Bohlmann, S., Heese, B., Hofer, J., Kanitz, T., Haarig, M., Ohneiser, K., Radenz, M., Seifert, P., Skupin, A., Yin, Z., Abdullaev, S. F., Komppula, M., Filioglou, M., Giannakaki, E., Stachlewska, I. S., Janicka, L., Bortoli, D., Marinou, E., Amiridis, V., Gialitaki, A., Mamouri, R.-E., Barja, B., and Wandinger, U.: DeLiAn – a growing collection of depolarization ratio, lidar ratio and Ångström exponent for different aerosol types and mixtures from ground-based lidar observations, Atmos. Meas. Tech., 16, 2353–2379, https://doi.org/10.5194/amt-16-2353-2023, 2023.

Burton, S. P., Hair, J. W., Kahnert, M., Ferrare, R. A., Hostetler, C. A., Cook, A. L., Harper, D. B., Berkoff, T. A., Seaman, S. T., Collins, J. E., Fenn, M. A., and Rogers, R. R.: Observations of the spectral dependence of linear particle depolarization ratio of aerosols using NASA Langley airborne High Spectral Resolution Lidar, Atmos. Chem. Phys., 15, 13453–13473, https://doi.org/10.5194/acp-15-13453-2015, 2015.

---

## Author Comment (AC2)

**Response to Referee #4**

**We thank the referee for carefully reading and reviewing this manuscript. Below are the original comments in italics with our responses in bold text.**

*The paper by* Floutsi et al. *introduces a novel aerosol classification method with high flexibility in terms of channel configuration. The output considers aerosol mixtures of four aerosol components that are directly linked to the lidar intensive properties. The authors provide the necessary information and guidance to understand the optimal estimation method, and, furthermore, to perceive the methodological ramifications through suitable visualization and specific case studies. The language and presentation are commendable throughout. The paper therefore is suitable for publication with only typographical corrections.*

*In the following, I included only a few minor comments and corrections that I hope will improve the manuscript.*

**Thank you for your positive feedback on our work as well as for helping us to improve our manuscript.**

*Minor Comments:*

*Ln117: I suppose that 30 iterations is an empirical estimate. How long does it take for the algorithm to provide output? Can it be used operationally?*

**Indeed, the choice of a maximum of 30 iterations as the convergence failure is made empirically and usually, the algorithm provides output within five iterations. Of course, this threshold does not prohibit the algorithm from being used operationally. For clarity, we have provided further justification for our choice of maximum iteration number in lines 126-128 of the revised manuscript.**

*Ln144-148: Do you apply quality screening to the input optical parameters? If yes, what are the threshold of acceptance for these parameters? Also, what does "order of appearance in the vector" mean ? Does it imply the discriminatory power of the intensive properties?*

**At the moment, there is no dedicated quality screening within HETEAC-Flex for the optical parameters, and we are currently completely relying on the quality assurance procedures applied to the input by the data providers, i.e., ourselves or ACTRIS (e.g., Bravo-Aranda et al., 2016; Wandinger et al., 2016b; Freudenthaler, 2016; Belegante et al., 2018; Freudenthaler et al., 2018). In particular, the PollyXT lidar systems (as well as their predecessors) follow the EARLINET (European Aerosol Research Lidar Network) standards.**

**No, the phrase "order of appearance in the vector" does not refer to the discriminatory power of the intensive properties, only to the order in which the input parameters must be compiled in the input file. As each element of the measurement vector is involved in different computations the order needs to be respected for the algorithm to work properly. Thus, it is just a technical prerequisite.**

*Ln236-239: Why the penalty term is not enough? Is the second criterion only invoked when the total relative contribution is greater than 1?*

**The penalty term covers only the scenario of a retrieved relative volume contribution per component being either below 0 or above 1. The other criterion (normalization) is invoked when**

the retrieved relative volume contribution of a component is smaller than 0 or indeed as you mentioned, when the total relative contribution is greater than 1.

*Ln320: It would be better to stress earlier that the input is layer-averaged values. Could it be possible to apply HETEAC-Flex to high temporal resolution lidar maps?*

Thank you for this suggestion. In fact, we had mentioned in line 144 of the original manuscript that the input is layer-averaged values. To emphasize this, we have slightly rephrased our statement (line 155 of the revised manuscript). In theory, yes applying HETEAC-Flex to high temporal resolution lidar data is possible but computationally expensive.

*Figure 7 and Ln370-371: Why color ratio and particle depolarization ratio do not have error bars? This is a bit confusing for me. Also, do you consider the standard deviation of the averaged intensive profiles for the error estimation?*

The data used in Fig. 7 have not been analyzed in this study, but in Tesche et al. 2011a, b, as indicated in the figure caption and in the text (lines 344-345 of the original manuscript). The absence of errors is also mentioned in lines 371-372, along with the errors that we assumed to enable the HETEAC-Flex retrievals. For the aforementioned reasons, we initially chose not to include the assumed errors in Fig. 7. Reviewer #3 raised a similar comment, and we have updated both Fig. 7 and the figure caption to include the errors of the color ratio and the particle linear depolarization ratio (with faint error bars).

*Ln354: Can you expand on how you defined the aerosol layers? Is layer detection part of the methodology?*

Layer detection is not part of the methodology, and, therefore, not mentioned in the manuscript. For the case studies presented in the paper we have identified the aerosol layers by manually inspecting the vertical profiles of the optical properties. We have included a relevant statement in lines 297-298 of the revised manuscript.

**References**

Bravo-Aranda, J. A., Belegante, L., Freudenthaler, V., Alados-Arboledas, L., Nicolae, D., Granados-Muñoz, M. J., Guerrero-Rascado, J. L., Amodeo, A., D'Amico, G., Engelmann, R., Pappalardo, G., Kokkalis, P., Mamouri, R., Papayannis, A., Navas-Guzmán, F., Olmo, F. J., Wandinger, U., Amato, F., and Haeffelin, M.: Assessment of lidar depolarization uncertainty by means of a polarimetric lidar simulator, Atmos. Meas. Tech., 9, 4935–4953, https://doi.org/10.5194/amt-9-4935-2016, 2016.

Wandinger, U., Freudenthaler, V., Baars, H., Amodeo, A., Engelmann, R., Mattis, I., Groß, S., Pappalardo, G., Giunta, A., D'Amico, G., Chaikovsky, A., Osipenko, F., Slesar, A., Nicolae, D., Belegante, L., Talianu, C., Serikov, I., Linné, H., Jansen, F., Apituley, A., Wilson, K. M., de Graaf, M., Trickl, T., Giehl, H., Adam, M., Comerón, A., Muñoz-Porcar, C., Rocadenbosch, F., Sicard, M., Tomás, S., Lange, D., Kumar, D., Pujadas, M., Molero, F., Fernández, A. J., Alados-Arboledas, L., Bravo-Aranda, J. A., Navas-Guzmán, F., Guerrero-Rascado, J. L., Granados-Muñoz, M. J., Preißler, J., Wagner, F., Gausa, M., Grigorov, I., Stoyanov, D., Iarlori, M., Rizi, V., Spinelli, N., Boselli, A., Wang, X., Lo Feudo, T., Perrone, M. R., De Tomasi, F., and Burlizzi, P.: EARLINET instrument intercomparison campaigns: overview on strategy and results, Atmos. Meas. Tech., 9, 1001–1023, https://doi.org/10.5194/amt-9-1001-2016, 2016b.

Freudenthaler, V.: About the effects of polarising optics on lidar signals and the Δ90 calibration, Atmos. Meas. Tech., 9, 4181–4255, https://doi.org/10.5194/amt-9-4181-2016, 2016.

Belegante, L., Bravo-Aranda, J. A., Freudenthaler, V., Nicolae, D., Nemuc, A., Ene, D., Alados-Arboledas, L., Amodeo, A., Pappalardo, G., D'Amico, G., Amato, F., Engelmann, R., Baars, H., Wandinger, U., Papayannis, A.,

Kokkalis, P., and Pereira, S. N.: Experimental techniques for the calibration of lidar depolarization channels in EARLINET, Atmos. Meas. Tech., 11, 1119–1141, https://doi.org/10.5194/amt-11-1119-2018, 2018.

Freudenthaler, V., Linné, H., Chaikovski, A., Rabus, D., and Groß, S.: EARLINET lidar quality assurance tools, Atmos. Meas. Tech. Discuss. [preprint], https://doi.org/10.5194/amt-2017-395, in review, 2018.

Tesche, M., Groß, S., Ansmann, A., Muller, D., Althausen, D., Freudenthaler, V., and Esselborn, M.: Profiling of Saharan dust and biomass-burning smoke with multiwavelength polarization Raman lidar at Cape Verde, Tellus Series B-Chemical and Physical Meteorology, 63, 649–676, https://doi.org/10.1111/j.1600-0889.2011.00548.x, 2011a.

Tesche, M., Müller, D., Groß, S., Ansmann, A., Althausen, D., Freudenthaler, V., Weinzierl, B., Veira, A., and Petzold, A.: Optical and microphysical properties of smoke over Cape Verde inferred from multiwavelength lidar measurements, Tellus B, 63, 677–694,715, https://doi.org/10.1111/j.1600-0889.2011.00549.x, 2011b.

---

## Author Comment (AC3)

**Response to Referee #3**

**We thank the referee for their review and valuable feedback, which helped us improve the quality of our manuscript. Below are the original comments in italics with our responses in bold text.**

*The paper by Floutsi et al. introduces a novel aerosol classification method based on lidar derived intensive properties. The main results of the study are of interest. However, authors should point out the novelty of their study also regarding the existing typing schemes. I recommend the publication of the manuscript after minor revisions, considering some general and specific issues detailed below in my review.*

**Thank you for your positive evaluation of our work as well as for helping us to improve our manuscript by proposing to add further important information and clarifications.**

*General comment*

*Why did authors not mention all the existing typing schemes based on lidar intensive parameters? The added value of their aerosol classification method should be pointed out.*

**In the manuscript, we did not mention extensively the existing typing schemes based on lidar-derived intensive parameters because the list is long and we aimed for a rather short introduction, focussed on space-borne lidars. Since HETEAC-Flex will be a supporting algorithm mainly for the ground-based EarthCARE cal/val efforts, we thought it would be appropriate to mention the CALIPSO typing scheme, as we did in lines 28-32 as well as the drawback of Aeolus with respect to aerosol typing in lines 43-46 of the original manuscript. Nevertheless, we realize the importance and added value that this new information would bring to the manuscript and have included a new paragraph in lines 93-101 of the revised manuscript.**

*Line 112. Authors state that: «the covariance matrix S_ describes the measurement errors». How are measurement errors are being calculated?*

**The aforementioned description in line 112 is rather generic. In fact, the actual contents of the measurement vector, along with the covariance matrix are discussed in Sec. 2.2. Now we have included a statement regarding the measurement error calculation in lines 163-164 of the revised manuscript.**

*Line 117. Authors state that: «Typically, the process converges within 30 iterations and if not, then it fails to converge and, consequently, there is no optimal solution.» How this value is being selected?*

**Usually, the algorithm provides a solution within five iterations, regardless of whether this optimal state is statistically significant or not. We, therefore, determined this number empirically as it is large enough to reflect that even if the process converges the solution will not be optimal nor statistically significant, as the cost function is most likely "trapped" in a local instead of the global minimum.**

**Reviewer #4 (RC2) raised a similar comment as well, and we have justified the choice of the maximum number of iterations in lines 126-128 of the revised manuscript to provide more clarity.**

*§3 Application of HETEAC-Flex. How are layers been defined? Authors should provide information on the layering detection.*

**Thank you for commenting on layer detection. Indeed, we had omitted to include information about it. For all cases, including the long-term dataset from Haifa, the aerosol layers have been identified**

manually. At the moment aerosol layer detection is not included in HETEAC-Flex and one should perform it separately and beforehand. This information is now included in lines 297-298 of the original manuscript. In the future, we would like to include aerosol layer detection in the HETEAC-Flex methodology, most likely following the wavelet covariance transform (WCT) method, similarly to EarthCARE (Wandinger et al., 2023b).

*Figure 7. Errors should be added in all products.*

We have updated both Fig. 7 and the figure caption to include the errors of the color ratio and the particle linear depolarization ratio (with faint error bars). Reviewer #4 had a comment related to the errors in Fig. 7 as well; please refer to our response (Reply on RC2) for further details.

*§3.3 has different structure than the 3.1 (Case, Overview and Aerosol characterization) and 3.2 (Case, Overview and Aerosol characterization). Paragraph 3.3 should be homogenized with the previous ones.*

Thank you for pointing out that the structure of Sect. 3.3 is not consistent with the previous sections. We have updated this part of the manuscript accordingly.

*Figure 6. Why are these retrievals modes selected? Authors could present all the available modes for this specific case study.*

You are right, all retrieval modes were applicable in this case. Therefore, we have updated Fig. 6 and we have revised accordingly the discussion in Sect. 3.1.2.

*Specific comments*

*Figure 8. Legend shouldn't be color filled.*

Indeed, Fig. 8 has been updated accordingly in the revised version of the manuscript.

**References**

Wandinger, U., Haarig, M., Baars, H., Donovan, D., and van Zadelhoff, G.-J.: Cloud top heights and aerosol layer properties from EarthCARE lidar observations: the A-CTH and A-ALD products, Atmos. Meas. Tech., 16, 4031–4052, https://doi.org/10.5194/amt-16-4031-2023, 2023b.